# Recent Process in Microrobots: From Propulsion to Swarming for Biomedical Applications

**DOI:** 10.3390/mi13091473

**Published:** 2022-09-05

**Authors:** Ruoxuan Wu, Yi Zhu, Xihang Cai, Sichen Wu, Lei Xu, Tingting Yu

**Affiliations:** Guangzhou International Campus, South China University of Technology, Guangzhou 511442, China

**Keywords:** micro/nanorobots, propulsion mechanism, swarming behavior, targeted delivery, in vivo imaging, minimally invasive surgery, image-guided therapy

## Abstract

Recently, robots have assisted and contributed to the biomedical field. Scaling down the size of robots to micro/nanoscale can increase the accuracy of targeted medications and decrease the danger of invasive operations in human surgery. Inspired by the motion pattern and collective behaviors of the tiny biological motors in nature, various kinds of sophisticated and programmable microrobots are fabricated with the ability for cargo delivery, bio-imaging, precise operation, etc. In this review, four types of propulsion—magnetically, acoustically, chemically/optically and hybrid driven—and their corresponding features have been outlined and categorized. In particular, the locomotion of these micro/nanorobots, as well as the requirement of biocompatibility, transportation efficiency, and controllable motion for applications in the complex human body environment should be considered. We discuss applications of different propulsion mechanisms in the biomedical field, list their individual benefits, and suggest their potential growth paths.

## 1. Introduction

Modern surgery is advancing towards the direction of non-invasive, mechanized and intelligent. Benefiting from the development of micro/nanotechnology, robotics and nanomedicine, micro/nanomachines that incorporate the advantages of multidisciplinary approaches have arisen. Different from traditional invasive manual operation, the micro/nanorobots can penetrate deep regions of the human body and cure diseases like a mini-doctor. Micro/nanorobots are supposed to navigate in a complex, highly viscous environment with driving energy supplied externally or by themselves [1]. As a micro medical device, these synthetic robots are fabricated in the form of spirals, rods, spheres, gears, and cells that move at microscales to complete their tasks under control [2,3,4,5]. They can swim to the targeted location and deliver the drug as well as serve for in vivo imaging [6,7,8,9,10,11,12,13,14,15,16,17].

For biomedical applications, the synthesis of micro/nanorobots integrating respective functions at the microscopic scale is essential. Achieving effective actuation, transportation and wireless manipulation on small scales has been a major challenge. Due to their unique size and dynamics, there are many micro/nanofluidic problems involved in the propulsion and application of swimming micro/nanorobots. Considering that micro/nanoscale objects in the movement are limited by Brownian motion as well as low Reynolds numbers (the ratio of inertial forces over viscous forces), designs based on classical mechanics are no longer applicable to micro/nanorobots. In addition, individual robot shows limited ability in executing tasks. As the need for biomedical application scenarios grows, micro/nanorobots are expected to perform in a cluster manner. There will be greater opportunities for application if the individual robot can self-assemble into cluster form and interact with each other to carry out specified tasks cooperatively.

Organisms in nature have inspired researchers with the solution to the above mentioned problems. Tiny single-celled microorganism can actively swim, sense their surroundings, and react to external stimuli [18]. Artificial micro/nanorobots have been developed by imitating the structure and propulsion principles of biological microswimmers. Replicating the structure design of microorganisms enables the artificial microrobots to propel [19,20,21,22]. Moreover, the exploration of a working principle in a collective manner of robots is noteworthy. Most individual organisms in nature can associate with each other and live in a swarm to perform biological activities efficiently. This collective working manner can be enlightened by the swarming intelligence in nature [23,24]. In recent years, much work has been done such as adding external fields and designing asymmetric structures to induce the cooperative swarming of microrobots for bio-imaging, cargo transportation, etc. [19,25,26,27,28,29].

In this review, we present an overview of recent progress in propelling and swarming methods of micro/nanorobots, highlighting its achievements in biomedical applications (Figure 1). In recent years there are some reviews summarized the basic propulsion mechanisms [30,31,32], some considered the mechanisms of field driven assembly (assembly of artificial micro/nanoparticles into multiscale structures) [33,34,35], and some reviewed the multiagent control (all microbots accept the same control inputs without involving interaction in their motion) [36,37]. This review focuses on how individual microbots work collaboratively to achieve biomedical applications. By showing recent research achievements, we illustrate the opportunities and challenges facing the practical application and provide perspectives for microrobots with higher biocompatibility, treating precision, and more application scenarios.

## 2. Principles of Main Driving Modes and Their Applications

This section is divided into 4 parts: magnetically, acoustically, chemically/optically and hybrid driven mechanism. Common hybrid driving modes such as co-mediated ultrasonic field and magnetic field, applied ultrasonic field and light illumination are listed below. This work aims to identify appropriate driving mechanisms and swarming strategies for biomedical applications. With the aim of establishing a framework of the recent achievements of the propelling mechanism and their biomedical applications of the microrobots, Table 1 summarizes the major breakthroughs in the field of microrobots under different propulsion methods in recent years, list important parameters such as materials, dimensions and speed of these particles. The following sections will describe these examples in detail.

### 2.1. External Magnetic Field Driven Mechanism

Magnetic actuation enables the controllable movement of magnetic micro/nanorobots under the modulation of magnetic fields. The magnetic field is a vector field that describes the magnetic influence on moving electric charges, electric currents, and magnetic materials. Currently, many different kinds of magnetic fields are utilized to propel the micro/nanorobots, including uniform magnetic fields rotating around the central axis, uniform oscillating magnetic fields, and pulsed magnetic fields varying with time and magnetic gradients in a specific direction. Among many dynamic fields, rotating fields (e.g., planar or conical fields) and oscillating fields in distinct directions are the two most common configurations. Rotating fields can be utilized to induce rotational motion [38,71]. In contrast, oscillating fields are widely adopted to activate traveling undulatory locomotion for some micro/nanorobots consisting of solid segments, soft tails, or hinges [72,73]. Besides, the dynamic magnetic fields can be heterogeneous, which can be manifested by field gradients, where the field strength varies with position. Magnetic objects will be subjected to magnetic forces in gradient fields, and the magnitude of the forces on the two ends of the object are different so that the net forces could drive the object in such fields [74,75]. 

As for the advantages of magnetic fields, the overall intensity of the magnetic field used to drive the magnetic micro/nanorobots is low, and it penetrates biological tissue through a low-frequency magnetic field. Hence, the damage to biosystem is low, guaranteeing the possibility of biomedical applications and excellent biocompatibility [76,77]. Meanwhile, the magnetism could wirelessly propel the micro/nanorobots in the local chemical environment and move the micro/nanorobots or swarms into the targeted position, indicating its crucial role in targeted delivery or therapy.

#### 2.1.1. Basic Propulsion Mechanisms

Micro/nanorobots impelled by magnetic fields always depend on the magnetic gradient and magnetic torque. The magnetic force and magnetic torque on a magnetic object in a magnetic field can be calculated from Equations (1) and (2):
(1)$$F=\left(m\cdot \nabla \right)\;B$$
(2)T=m ×B
where F and T are the magnetic field gradient force and magnetic field torque, respectively, on a magnetic object at a point in a magnetic field; B is the magnetic flux density at a point, and m is the magnetic dipole moment of the magnetic object; The torque per volume aligns an object with magnetization M with the applied field B.

From Equations (1) and (2), it is obtained that when the magnetic field is uniform, the magnetic force on the micro/nanorobots as magnetic dipoles is zero. If the direction of the uniform magnetic field is the same as the direction of the magnetic dipole moment of the magnetic object, the magnetic torque is also zero. Only if the direction of the magnetic dipole moment is not co-linear with the direction of the applied magnetic field, the magnetic object would be subjected to the magnetic torque and deflected in the direction of the applied magnetic field. The motion does not stop until the direction of the magnetic dipole moment is co-linear with the direction of the applied magnetic field. Inspired by these phenomena, researchers can control the start and stop of the movement of magnetic objects by changing the magnetic field. Therefore, the presence of a varying magnetic field spatiotemporally is an indispensable condition for driving a micro/nanorobot into continuous motion. 

Purcell’s Scallop Theorem states that the inertia effect can be ignored in the environment at a low Reynolds number, and the reciprocating motion of particles cannot make them advance. So micro/nanorobots driven by magnetic fields often require special designs to break the reciprocal motion. To begin with, it is representative and widely employed to manipulate micro/nanorobots with an asymmetric shape structure [78,79,80,81,82,83,84,85,86,87]. The second strategy is designing the flexible component, such as mimicking the flagellum of bacteria to generate spiral motion in the rotating fields or mimicking the flagellum of eukaryotic cells to produce flagellation in the oscillating fields. Third, the asymmetric assignment of magnetic fields can break the boundary conditions of symmetry, such as magnetic gradient fields where materials with magnetic properties are propelled by magnetic forces and move in the direction of greater magnetic field strength. And the fourth strategy focuses on the interaction between the micro/nanorobots and the surface of the adjacent object to actuate the robots to roll on the surface. All the strategies are aimed at breaking symmetry in the movement of micro/nanorobots, which could achieve by the novel functionalization design of its structure, constitution, or surface [37]. 

Magnetically driven micro/nanorobots operate in an environment of low Reynolds number (e.g., blood, lymphatic fluid, vesicular fluid, etc.), which imposes stringent requirements on the design characteristics of these robots [88]. Microorganisms usually live in low Reynolds number environments, and their actuation method enlightened the design of micro/nanorobots. For example, some bacteria have a molecular motor in their bodies that controls the helical rotation of the flagellum, which actuate the movement of the bacteria. To balance the generated torque, some bacteria control the helical rotation of the flagellum so that the flagellum rotates in opposite directions at two distinct frequencies, which actuate the movement of the bacteria (Figure 2a) [89]. Based on this actuation method of bacteria, researchers designed corkscrew-like micro/nanomachines called artificial bacterial flagella (ABF). An external rotating magnetic field plays an impotent role in powering the ABF system. And the steady rotation around its helical axis can be effectively converted into a translational motion whose direction is parallel to the rotation axis of the two-dimensional planar rotational field [90,91]. Magnetic field-activated micro/nanorobots were initially developed from helical microrobots. Honda et al. [92] and Kikuchi et al. [93] focused on the helical micro/nanorobots and designed centimeter-scale and millimeter-scale spiral robots, respectively. Under the action of the uniform rotating magnetic field, the tiny robot can move freely at a low Reynolds number. The researchers further reduced the size of the machine based on these designs. Nelson’s group prepared a micron-scale helical microrobot by self-scrolling technique [90]. Fischer et al. used the glancing angle deposition (GLAD) technique to design a nano-scale helical microrobot [78]. Under the rotating fields, there are many different kinds of propellers. ABFs composed of Cr/Ni/Au stacked films with helical tails move in water by wireless control of a low-intensity rotating magnetic field [94]. Nelson et al. designed the helical micromachines with different shapes manufactured by three-dimensional direct laser writing (DLW) and physical vapor deposition [21]. At a relatively low frequency, the micromachines wobbled about their helical axes. When the rotating field’s input frequency is high enough, these devices always generate corkscrew motion in the fluid. Gao et al. designed a microswimmer based on a helical plant vessel, which was made by simply coating a thin magnetic layer on the separated helical xylem vessel plant fiber [95]. Under the rotating magnetic field, the magnetic layer allows high-speed propulsion (over 250 μm/s) and precise direction control. 

In nature, some microorganisms swim at a low Reynolds number by the periodic movement of their cilia or flagella, called traveling-wave motion. Based on the example from nature, the researchers have used oscillating magnetic fields to simulate this movement and manipulated special micro/nanorobot type flexibly. This type of microrobot contains two primary components: a magnetic head and a flexible tail. The magnetic head moves back and forth under the influence of the magnetic torque when the direction of the applied uniform magnetic field oscillates at an angle. This causes the tail to paddle like a flexible oar, pushing the fluid to produce waves in order to propel the system ahead. It is noteworthy that such micro/nanorobots require an appropriate tail design. The oscillation of the magnetic head corresponds to a symmetric reciprocal motion when the tail is excessively short and inflexible. At a low Reynolds number, it is unable to achieve effective displacement. Additionally, when the flexible tail is excessively long, the robot experiences greater fluid resistance and less effective mobility.

Earlier, flexible micro/nanorobots driven by oscillating magnetic fields at the millimeter level were prepared by Guo et al. [99] and Sudo et al. [100]. Experimental results showed that these robots achieved significant motion in the water and viscous glycerol respectively by adjusting the magnetic field frequency, amplitude, and flexible tail parameters. The multilink nanoswimmer consists of a polymer and two magnetic metal nanowire links, which are connected by a hinge (Figure 2b) [96]. The planar oscillating magnetic field could react with the magnetic material Ni/Au to propel the motion of nanoswimmers. A magnetic artificial sperm microrobot consisting of a magnetic ellipsoid head and a flexible tail was prepared by Khalil et al. [101]. The robot is able to achieve a motion speed of 158 µm/s at 45 Hz. Li et al. designed an efficient bionic magnetic nanoswimmer with a deformable body, which successfully simulated the propulsion of the body and caudal fin displayed by fish (Figure 2c) [72]. A novel multilink two-arm artificial nanoswimmer was proposed by Li et al., which is made up of two nickel arm segments joined by a flexible porous silver hinge to a central gold body section [102]. Notably, the two-armed nanorobot demonstrates effective nonplanar locomotion in which the two arms swing together to move the center gold link forward in response to a planar oscillating magnetic field.

Surface rolling micro/nanorobots are robots that roll on the surface under the interaction between the magnetic field and the surface of adjacent objects. The introduction of physical boundaries to break spatial symmetry is a new strategy for actuating micro/nanorobots at low Reynolds number [100,101,102,103,104,105]. The characteristic of surface rolling micro/nanorobots was determined by the confining boundary (slip or no-slip), the distance between the objects and surface, and the frequency of magnetic fields (at a specific frequency, the micro/nanorobots have the fastest speed). In 2011, Mahoney et al. designed a surface rolling robot [106]. They loaded a cylindrical NdFeB magnet onto a motor near the surface where the magnetic ball was placed, adjusted the position of the motor, and rotated the motor at a specific frequency. The magnetic ball could keep rolling forward on the surface. Zhang et al. designed Ni nanowire surface rolling microrobots [97]. When different boundary conditions exist at the two ends of the nanowire, such as when the nanowire is close to a wall, a tumbling motion can be generated, resulting in the advancement of the nanowire. It was found that the nanowires advance with a tumbling motion despite the gap between the rotating nanowires and the vertical wall, indicating that the advancement is a result of the vertical boundary. If the nanowire is in a rotating magnetic field applied on a horizontal plane, the nanowire rotates in this plane due to the induced magnetic torque. In addition, the direction of nanowire propulsion can be reversed by reversing the direction of rotation of the magnetic field. It is known that the drag coefficient of an object increases as it approaches a wall. Therefore, if the nanowire rotates with an angular velocity of ω, the hydrodynamic interaction between the nanowire and the wall is stronger at the one side end than at the other side end, resulting in a difference in velocity between the two ends of the nanowire. Due to the dynamic boundary conditions, the center of rotation of the nanowire changes during the tumbling process (Figure 2d). 

#### 2.1.2. Swarming Behavior

Due to the limitations of the small size of the micro/nanorobot, it is challenging to obtain great drug-carrying capacity, motility, and signal feedback for medical imaging. The use of micro/nanorobotic cluster technology can provide practical solutions to these problems. Swarming or collective behavior has inspired researchers to design synthetic micro/nanomachines, which can perform collaborative tasks that can’t be accomplished with a single micro/nanorobot and cooperate to achieve more complex biological or environmental tasks [107]. Magnetic fields are an effective strategy for driving clusters of colloidal particles as low-intensity magnetic fields are readily available in a laboratory environment and can be used to control particle motion in three dimensions. 

Ferromagnetic (e.g., Ni, Co, Ni) and paramagnetic materials are always introduced to trigger the swarming behavior and achieve specific tasks by applied magnetic fields. During the formation of a swarm, a magnetic force is formed between two adjacent particles, and each particle is affected by other particles in its vicinity, resulting in the gradual construction of a collective behavior [108]. The magnetic force between two magnetic particles can be calculated from Equation (3):(3)$$f_{\mathrm{ij}}=\frac{3\mu_{0}}{4\pi ∥r_{\mathrm{ij}}{∥}^{4}}\left(\left({\hat{r}}_{\mathrm{ij}}^{T}m_{j}\right)m_{i}+\left({\hat{r}}_{\mathrm{ij}}^{T}m_{i}\right)m_{j}+\left(m_{i}^{T}m_{j}-5\left({\hat{r}}_{\mathrm{ij}}^{T}m_{i}\right)\left({\hat{r}}_{\mathrm{ij}}^{T}m_{j}\right)\right){\hat{r}}_{\mathrm{ij}}\right)$$
where $\mu_{0}$ is the permeability of free space and two particles are simplified as two magnetic dipoles ($m_{i}$, $m_{j}$ located at $P_{i}$ and $P_{j}$, respectively), $r_{\mathrm{ij}}={\;P}_{i}-P_{j}$. Besides, $\hat{r}$ is the unit vector between the two dipoles.

Each particle in a system composed of magnetic particles and a magnetic field will be driven by both the applied magnetic field and the magnetic moments of the adjacent particles. The mobility of magnetic particles disturbs the surrounding fluid or distorts the driving plane’s interface, which significantly impacts the motion of other agent particles. These magnetic dipole-dipole interactions and hydrodynamic interactions are essential to the evolution of collective behavior in magnetic particle and magnetic field systems. 

Earlier, in a system of rolling ferromagnetic microparticles powered by a vertically alternating magnetic field, Kaiser et al. reported the onset of flocking and global rotation. By combining experiments and discrete particle simulations, they identified the fundamental physical mechanisms that lead to the emergence of large-scale collective motion: spontaneous symmetry breaking of the clockwise/counterclockwise particle rotation, collisional alignment of particle velocities, and random particle reorientations due to shape imperfections [109]. In certain systems of particles and magnetic fields systems, the clustering behavior between particles caused by magnetic dipole interactions will play a decisive role. According to Yu et al., paramagnetic nanoparticles may be arranged into a ribbon-like dynamic microswarm under the actuation of oscillating magnetic fields [27]. Actuated by an oscillating field consisting of a superposition of a constant and an alternating field, dipole attraction between the induced moments exists. Following the first formation of the nanoparticle chain, its time-dependent orientation and length can be estimated using the torque balance-based method, which is related to the direction, intensity and angular velocity of the magnetic field. The magnetic attraction between chains causes them to dynamically fragment and reassemble, which results in pattern formation. The input parameters, such as the intensity and frequency of the rotating field, have an impact on the interaction between the magnetic chains. In addition, the field ratio, defined as the ratio between the alternating and constant fields, can also affect the collective state of a nanoparticle swarm. The dynamic decomposition and reconfiguration of the particle chains lead to the reversible elongation and contraction of the swarm pattern. The microswarm can perform reversible anisotropic deformation, controlled splitting and merging of subswarms. Advanced magnetically driven swarm control strategies are implemented by such magnetic dipole interactions resulting in controllable pattern transformation. 

In addition, there are also studies on the collective behavior of bionic micro/nanorobot swarms. The self-assembly of human artificial cilia is studied by Vilfan et al., a structure similar to artificial flagella that drive individual particles, demonstrating the feasibility of artificial cilia to generate fluid flow and serve as a transport platform [110]. Xie et al. have developed a magnetic peanut-like hematite colloidal microrobot, which is 3 microns long and 2 microns in diameter [25]. Colloidal particles can be operated to exhibit a range of motion modes, such as oscillating motion and a liquid-like state. When the rotating magnetic field is supplied to the (x, z) and (x, y) planes, the particles exhibit rolling and rotating motion modes, respectively. The colloidal particles rolling along the short axis accumulate and assemble to create a chain swarm that rolls in the direction parallel to the short axis owing to the interactions of magnetic dipoles and fluid fields around the individual. Besides, a vortex flow field forms due to the colloidal particles rotating with the long axis as the center in the rotating magnetic field of (x, y) plane. The vortex-induced hydrodynamic interaction and magnetic dipole interaction trigger the adjacent colloidal particles constantly assemble to form vortex-like swarms. Under the conical magnetic field, colloidal particles display a tumbling motion that is principally produced by the permanent torque induced by the *X*-axis component of the conical magnetic field, which promotes the spontaneous construction of a chain swarm along the magnetization direction. By altering the magnetic field mode and frequency, the transformation and movement of the swarm among the four kinds of swarms can be accurately controlled, and reversible reconfiguration of the swarms is possible. Because the microrobot is tiny enough, it can reach places that are difficult to treat directly by other means, such as the extremities of capillaries and the retina. The swarm can demonstrate a range of modalities that respond swiftly to the environment or task under the modification of a rotating magnetic field. It can morph into a long chain modality to easily penetrate tight simulated capillaries and is also capable of mimicking a genuine ant colony (vortex modality) and a herring predator array (crossband modality), respectively (Figure 2e). Massana-Cid et al. assembled colloidal carpets using paramagnetic colloids with a radius of 1.4 μm [98]. The paramagnetic particles could self-assemble into a circular rotating pattern under a rotating magnetic field. Initially, the particles were subjected to gravity and electrostatic force, so they kept a balance in the water. Due to a rotating field in the plane (x, y), which is expressed as:(4)$$B\left(t\right)=B\left[\cos\left(2\pi \mathrm{ft}\right)\hat{x}-\sin\left(2\pi \mathrm{ft}\right)\hat{y}\right]$$
where B is the field strength and f is the frequency and attractive dipolar interactions, the particles formed a compact cluster. When the direction of $m_{i}$ and $m_{j}$ are parallel to $r_{\mathrm{ij}}$, the magnetic force between two dipoles has a maximum value. By inducing a rotating magnetic field in the plane (x, z) and an oscillating component along the 𝑦 direction, the particles could move in a specific direction. When the external field is removed, the dipole-dipole interactions vanish, resulting in the dissociation of the formed pattern (Figure 2f). 

The creation of microswarms triggered by the interaction between the particles and the surrounding dynamic fluid is another fundamental mechanism to explain the generation of collective behavior and also an essential factor in the study of the dynamical behavior of magnetic particles. Yu et al. reported the generation and navigated motion of a vortex-like paramagnetic nanoparticle swarm [111]. By adjusting the actuation parameters, they can perform pattern reconfiguration to accommodate complex and convoluted environments, which facilitates more accurate targeted delivery. The nanoparticle chains are capable of rotating under a horizontal rotating magnetic field, which disturbs the fluid near the bottom boundary and generates many coaxial vortices. Stronger inward forces can be produced by the flow around the swarm that has a higher velocity, which causes the pattern of swarm structure to become more compact. When the particle chain rotates in the direction of the long axis of the elliptical field, long chains are formed and bring high flow velocities. The cluster pattern contracts in that direction as a consequence of the increased confinement force. In contrast, when the particle chain rotates in the direction of the short axis of the elliptical field, the chain tends to disassemble into shorter chains due to the lower field strength.

If the magnetic dipole connection between the particles and the hydrodynamic interaction between the particles and their environment are too weak to effect the particles’ motion, then the particles travel practically independently under the influence of an applied magnetic field. For example, in a paramagnetic nanoparticle-based system, a uniform magnetic field is applied in the *z*-axis, and a rotating magnetic field is applied in the x-y plane to create repulsive interaction forces on the magnetic nanoparticles [112]. The chain has unequal drag pressures at the top and bottom during rotation as a result of the differing distances from the two ends to the boundary, causes the chain to tumble along the boundary. Reducing magnetic and hydrodynamic interactions by increasing the distance of the decomposed chains from each other, which leads to less chance of recombination and makes the decomposition process more stable. Under the effect of magnetically induced repulsive forces, these shortened chains exhibit motion spreading in different directions and thus have little chance to interfere with each other.

#### 2.1.3. Biomedical Application

With the advantages of easy adjustment, excellent penetrability, and harmlessness to biological tissues among all actuation strategies, the micro/nanorobots associated with magnetic fields have many applications, especially in biomedicine [89]. Magnetically propelled micro/nanorobots with controllable motility, access to tiny lumens, and swarming enhancing effects are capable of physically destroying damaging biological structures without inducing medication resistance. The main applications of micro/nanorobots driven by a magnetic field are targeted drug delivery, minimally invasive surgery, cell manipulation, biopsy, biofilm disruption, imaging-guided delivery/therapy/surgery, biosensing, etc. 

Through the interaction of magnetic fields, drug delivery in cells by collective micro/nanorobots is an exciting potential application, which requires in-depth study of single-cell drilling and intracellular movement. The motile metal-organic frameworks (MOFs) material is a micro-scale robotic platform for medical applications, such as targeted drug delivery and nanoscale surgery. Wang et al. have successfully prepared a zinc-based MOF (ZIF-8) magnetic helical structure with good biocompatibility and pH-responsive properties, and the microswimmer could swim along the expected trajectory under the weak rotating magnetic field [38]. The results demonstrated that this system could achieve single-cell targeting in cell culture media and controllable cargo delivery in complex conditions. Lee et al. used an electromagnetic actuation (EMA) system to develop therapeutic drug delivery microrobots for precise targeting [39]. As shown in Figure 3a, their team proposed a biocompatible and hydrolyzable PEGDA-based helical microrobot containing the anticancer drug doxorubicin (DOX), which can be delivered to the lesion of cancer cells under magnetic fields. Park et al. developed a degradable hyperthermia microrobot (DHM) with a 3D helical structure, which can be used for drug delivery, release, and hyperthermia therapy [113]. The microrobot is controlled remotely by a rotating magnetic field with high precision. By a hyperthermic effect, drug-free DHMs reduce cancer cells’ viability by elevating the temperature under an alternating magnetic field. Similar to the helical propulsion micro/nanorobots using bioinspired methods, a biomimetic magnetic microrobot (BMM) inspired by magnetotactic bacteria (MTB) is designed to apply to targeted thrombolysis (Figure 3b) [114]. Due to the aligned iron-rich nanocrystals in MTB, the bacteria can move under the magnetic fields [115]. The BMM is composed of linearly aligned iron oxide nanoparticle (a particle belonging to MNPs) chains, which are achieved due to the interparticle dipolar interactions of MNPs under a static magnetic field. There is a promising application for ultra-minimal invasive thrombolysis with the delivery and release of thrombolytic via the swarm of BMMs. Yu et al. implemented an ex vivo generation of medium-induced swarms and their targeted delivery in a bovine eyeball [116]. The generation process and navigated locomotion process in vitreous humor are shown in Figure 3c. They utilize a Helmholtz electromagnetic coil as the propulsion setup with an ultrasound transducer. First, concentrated particle suspension is injected into the eyeball, and then, the collective behavior of the nanoparticles is triggered by the rotating magnetic fields. These efforts clarify the basic understanding of the microrobot population, which is an essential step toward targeted delivery in vivo. The swimming nanorobot, which can move controllably in the vitreous of the eye, tries to break through the biological barrier and complete the active drug transportation by relying on the autonomous movement of the nanorobot under the magnetic field [117]. Inspired by the liquid lubrication interface of the pitcher plant, the researcher proposed a helical magnetic nanorobot with a nano-liquid lubrication layer on its surface, and its diameter was only 500 nm. Under the guidance of an external magnetic field, these helical nanorobots with surface lubrication can effectively overcome the adhesion of biomolecules and complete the long-distance controllable cluster movement from the center of the eye’s vitreous body to the retina, and reach the designated position in the vitreous body of the eye. In the future, these magnetic helical nanorobots with surface lubrication are expected to load drugs, get the focus by autonomous movement, perform active targeted drug delivery tasks, and realize the minimally invasive and precise treatment of diseases (Figure 3d). FePt, as a promising magnetic material for biomedical applications with excellent magnetic properties, especially for micro and nanodevices, can manufacture nanorobots and active cell targeting and magnetic transfection of lung carcinoma cells [118]. After internalization, cancer cells express an enhanced green fluorescent protein, and cell vitality is not affected by the FePt nano-propeller (Figure 3e). Owing to the large distance between the microrobots, neither magnetic nor hydrodynamic interactions are sufficient to affect the motion behavior of the adjacent microrobots. Independent motion and simple direction control are the most essential characteristics of weak interaction forces [37]. Cheng et al. utilized collective nanorods to enhance the mass transport of tissue plasminogen activator molecules at the blood clot interface [119]. Embolization is a clinical technique used to block vascular blood flow to treat tumors, fistulas, and arteriovenous malformations. Law et al. develop microswarms for embolization therapy. This work achieves selective regional embolization by controlling the motion and aggregation of magnetic particle clusters to reduce the risk of complications associated with existing embolization techniques. Through surface functionalization, magnetic particles containing thrombin as an embolic mediator are used to facilitate thrombus formation [120]. 

There is a bioinspired example of collective micro/nanorobots associated with viscoelastic fire ant aggregations. The researchers trigger ferrofluid droplets into microswarms by spatiotemporal programming applied magnetic fields [121]. The reconfigurability of microswarms adopts an environment of different types, and it is like a soft robot that could grasp a targeted object. The ferrofluid droplets immediately gathered under the rotating magnetic field into local dynamic clusters, and the microrobot swarm gradually formed. A low-frequency circular rotating magnetic field generates the round liquid microswarm (RLM). Under the high-frequency rotating magnetic field, the microswarm also experienced the transformation from liquid behavior to solid order. RLM and fusiform liquid-like microswarm (FLM) were transformed into a round solid-like microswarm (RSM) and a fusiform solid-like microswarm (FSM), respectively. It is demonstrated that the microswarm can navigate multiple terrains by changing the propulsion mode controlled only by a magnetic field. The RLM can pass through the uneven substrate, the wide semicircular groove, and narrow groove arrays. Also, the RSM can pass through the rough substrate and climb high stairs (Figure 3f). The microswarm is also suitable for grabbing a targeted object smoothly in a limited environment due to its excellent deformation ability, flexibility, and adaptability. The transformation between different microswarm and collective behavior patterns enables microswarm to complete various operational tasks and create potential solutions for micro-manipulation, micro-processing, and even biomedical applications, such as removing thrombus from blood vessels. Xie et al. developed an intelligent sea urchin-like micro-swimmer based on sunflower pollen grain (SPG), whose function is cell perforation and targeted drug delivery in Figure 3g [122]. After acidolysis, sputtering, and loading, the natural SPG can be placed in different magnetic fields to promote the particles’ motion (rolling and spinning) for cargo transportation and cell perforation. In addition, the assembly behavior between microperforators can further improve individual sports performance and adaptability. As shown in Figure 3h, navigating a multitude of helical microrobots in the intraperitoneal cavity of a Balb-C mouse under the whole-body fluorescence imaging [19]. It is a significant milestone for a swarm of magnetic helical microswimmers by external magnetic fields (less than 10 mT) in deep tissue in vivo imaging and actuation. Sun et al. explored the method of biofilm eradication from medical implants [123]. Their team used natural sunflower pollen to design the magnetic urchin-like capsule robots (MUCRs) loaded with magnetic liquid metal droplets (MLMDs, antibacterial agents). It was discovered that an applied magnetic field stimulates the appearance of swarms and induces the transformation of MLMDs into spherical and rod-like structures with sharp edges. As for the mechanism of biofilm eradication, the swarms would generate a mechanical force due to the microspikes of MUCRs and the sharp edges of MLMDs when disrupting biological matrices and bacterial cells. Through endoscopy, the studies indicated that the microswarm could reach the biliary stent accurately and in ten minutes. In addition, the researchers were able to study fluoroscopic images for real-time delivery and navigation of the microswarm. Thrombosis is a common intravascular disease with multiple clinical manifestations and complications. Traditional treatment protocols remove thrombus by injectable thrombolytic drugs or catheter interventional techniques, but thrombolytic drugs lack targeting, and catheter interventional techniques require high operator experience and judgment, and improper operation can easily damage the blood vessel (Figure 3i). Wang et al. proposed a small-scale robot-based scheme for thrombus localization and accelerated thrombolysis [124]. The helical-shaped microrobot was fabricated using a 3D printing process and automated delivery using a dynamic magnetic field, while ultrasound imaging was used for real-time robot localization and environmental monitoring. The robot was able to locate the thrombus location in real time and accelerate the thrombus lysis. Experimental results showed that the helical robot exhibited excellent structural stability in the blood environment and blood flow environment and was able to maintain the completed overall structure and be recovered after the thrombus lysis task was completed. By monitoring the robot’s motion in real time and the Doppler ultrasound signal induced by the robot, the researchers successfully localized the thrombus blockage site in a complex dynamic environment of similar blood vessels. The microrobot, driven by a magnetic field, was able to generate strong convection to accelerate the material exchange of thrombolytic factors while exerting shear stress on the blood-thrombus interface to facilitate the removal of thrombolytic products. The magnetic robots consisting of magnetic materials would generate reactive oxygen species (ROS) in vivo and cause tissue/organ damage via Fenton/Fenton-like reactions (Figure 3j). Zhao et al. designed a magnetically driven ROS-scavenging nanorobots (ROSrobots) [125]. They utilized ferromagnetic elements of microrobots to react with [Fe(CN)_6_]^4−^ ions to generate Prussian blue in situ on the surface of nanorobots, which overcomes the problem of magnetic material-mediated Fenton reaction. Their group also constructed reconfigurable ROSrobot swarms capable of multimodal transformation, motion, and manipulation via an ultrasound imaging-guided electromagnetic drive system. In addition, their research focused on the ROSrobot swarm in a joint cavity of a bone isolated from rats for the active targeting of osteoarthritis and demonstrates the possible in vivo and clinical applications of magnetic nanorobots (Figure 3k).

### 2.2. External Acoustic Field Driven Mechanism

Ultrasound, with a frequency above the threshold of human ear discrimination, is a well-directed, penetrating and less harmful energy source. The acoustic driven method uses acoustic waves generated by an external field device to provide driving force for particles suspended in solution. The acoustic field-driven method can propel particles, assemble them into a specific shape and serve for other applications that will be mentioned in this paper later.

Many theories and methods can be used to explain the process of converting the energy of ultrasound into other forms of useful work. Among them, in order to convert high-frequency ultrasound waves directly into stable forces on objects, it is necessary to use nonlinear acoustic mechanisms, which include acoustic streaming forces and acoustic radiation forces. Acoustic streaming transfers energy by the momentum transferred by fluid flow. If acoustic streaming is present, particles will receive a drag force. The viscous stress (T) is related to the gradient of viscosity (μ) and velocity(v), which satisfy the relationship [126]:(5)$$T\;=\;\mu \nabla \vec{v}$$

When the wavelength of propelled particles is significantly smaller than the acoustic wavelength, acoustic radiation forces (ARF) can be employed to generate steady forces. Particles with a homogeneous spherical shape will receive ARF based on the parameters of the acoustic field and the particles themselves. The force follows the equation: (6)$$F_{\mathrm{ARF}}=-\nabla U_{\mathrm{ARF}}$$

That is, the force equals the gradient of a potential. There is also a proportional relationship here:(7)$$U_{\mathrm{ARF}}\;\propto \;I\Phi V$$
where I represents acoustic intensity, Φ represents acoustic contrast factor of particles, and V the volume of particles. 

Bubbles are a special type of acoustic material often encapsulated in a viscoelastic shell or embedded in a medium. When ARF is applied to bubbles in an acoustic field, the bubble will receive both primary and secondary Bjerknes forces. The primary Bjerknes force can be given by:(8)$${\vec{F}}_{B1}=-V\nabla P$$
where V represents the volume of bubble, ∇P represents the gradient of acoustic pressure amplitude, and this force pushes bubbles toward regions of high pressure. The secondary Bjerknes force has an effect on each bubble. Depending on the bubble’s size and the driving frequency, this force can attract or repel bubbles. By varying the size and driving frequency of the bubbles, the force can attract or repel bubbles, controlling the particles motion.

#### 2.2.1. Basic Propulsion Mechanisms

Acoustic streaming is a simple method for propelling particles in an acoustic field. As shown in Figure 4a, under the influence of acoustic streaming and radiation force, the acoustic field can cause carbon nanotubes (CNTs) on solid substrates to be aligned in parallel [127]. A similar phenomenon was discovered by Oberti’s [128] and Shi’s [129] teams, by which the assembly of particles can be controlled into distinct patterns. In these directed propulsion phenomena, particles are initially propelled by the time-averaged pressure gradient of the acoustic field, which imparted a force toward or away from the pressure nodes. The particles are subsequently dragged along with the streaming flow generated by the geometry of the fluid cell. 

Under the influence of acoustic streaming and radiation force in an acoustic field environment, asymmetrically structured particles can generate a propulsive force. In Wang’s study [46], instead of being dragged by the flow induced by the acoustic field, the microrod can move towards a pressure node (Figure 4b). The speed of motion can be adjusted by varying the amplitude of the ultrasound waves, and the asymmetric structure of the metal nanorod allows it to undergo directional motion. Under the influence of continuous or pulsed ultrasound, the metal nanorods can move at approximately 200 μm/s, with one end of the rod (Ru or Pt) always facing forward. Utilizing this property makes the manipulation of the nanorods possible. Soto et al. demonstrated an efficient design of a nanoshell (Figure 4c) [47]. The driving force in an ambient acoustic field applied on the nanoshell consists of the ambient fluid flow and the acoustic radiation forces since the total streaming pressure does not cancel out due to its asymmetric structure.

Under acoustic actuation, the microrobot was propelled by the synergistic effect of ARF generated by the scattering of acoustic waves and the streaming of bubbles. Feng et al. designed a microswimmer powered by oscillating the trapped gaseous bubble, and achieved bidirectional propelling movement [48,132]. An external acoustic wave will stimulate the bubble, leading to microflows that propel the microswimmers. The difference between the intake and discharge flows dictates the magnitude of driving force, and as the size of the microswimmer decreases, the Reynolds number also lowers, resulting in a loss of efficiency. To maintain the Reynolds number, the amplitude and frequency of bubble oscillation must be sufficient to generate a powerful propulsion force. Aghakhani et al. proposed a unidirectional swimming bullet-shaped microbot fabricated by 3D printing [50]. A spherical air bubble resonating with acoustic waves was contained within its body cavity. A little fin was added to the surface of this special-shaped particle to assure unidirectional motion, and a soft magnetic nanofilm layer was incorporated for direction control. In particular, the microrobot’s speed can reach up to 90 body lengths per second, or 2250 μm/s. When the microrobots were immersed in a fluid medium with the acoustic field, the acoustic pressure was generated and mapped (Figure 4d). The incorporation of an oscillating tail structure into the design of a micro/nanorobot is a distinct approach for propelling particles, allowing for mobility with a large amplitude of propulsion. Ahmed et al. developed a nanorobot composed of an acoustically active flexible tail and a bimetallic head of Au/Ni (Figure 4e) [51]. In an acoustic field with a wavelength much larger than the swimmer dimension, the PPy tail began to oscillate and generated a steady streaming flow. With this kind of field called first-order fields and the second-order steady flow caused by the first-order fields, nanorobots experience a Reynolds stress, which can propel the nanorobot and determine its direction. Ren et al. designed a microswimmer platform based on acoustic bubbles [133]. The combination of secondary Bjerknes force and streaming propulsive force enables this microrobot to move forward in the acoustic field. When an external magnetic field is used to break the balance between these two forces, the modulation of motion direction and speed of the microrobot at the 3-D boundary or in free space can be achieved. The strong propulsive force and precise directional control enable the microswimmers to operate robustly and reliably in complex environments.

In an acoustic hologram developed by Melde et al., particles can be assembled into arbitrary shape by properly designing the acoustic pressure field, such as controlling the attenuation of the sound pressure field. The measured acoustic pressure in the image plane is shown in Figure 4f [130,134,135]. Radiation force is used to propel particles to the nodes or antinodes of the standing acoustic field. Acoustic streaming flows lead to recirculating convection fields, which transport those suspended cells to target place for further assembly. The streaming-induced acoustic holography technique can be used to construct complex cell patterning, which can can contribute to bionic applications of complex cellular arrangements.

#### 2.2.2. Swarming Behavior

The growth of functions and applications of ultrasound-induced assembly is assisted by the participation of more particles in the assembly process, which results in the formation of unions or patterns. This section introduces the assembly behaviors of microparticles in an acoustic field.

By incorporating magnetic materials in the fabrication of microrobots, their motion modes are changed by the synergistic effect of magnetic forces in the acoustic field. Ahmed et al. found that segmented gold-ruthenium nanorods with thin Ni segments at one end can be assembled into geometrical particle union (Figure 4g) [131]. Under the fluids by excitation with about 4 MHz ultrasound, interactions between the magnetic Ni segments happen and generate the assembly of multimers which exhibit several distinct modes of motion. The density and speed of the particles, which are dependent on the ultrasonic power, determine the number of monomers in multimers. After removing the acoustic field, this kind of union may still be propelled by an external magnetic field.

In the study of Li et al., a novel micromotor with concave structure was designed to achieve the attractive group movement behavior of the motor, including aggregation, group migration and dispersion, by adjusting the sound field [136].

#### 2.2.3. Biomedical Applications

Numerous uses for acoustic swimmers that exhibit various propelling and swarming behaviors are available. Acoustic swimmer applications include patterning and assembly of biological materials, in vivo imaging and drug delivery will be covered in this section. Researchers focused on the strategy of using acoustic fields to assemble cells and pat-tern shapes as being benign to the cells. For hybrid smart materials, cells are appropriate to be used for creating unions. Most acoustic methods used in patterning and assembling cells can be clarified into three categories, standing wave trapping [137,138], Faraday wave patterning [139,140,141] and holographic patterning [130,142]. Using the aforementioned assembly techniques, cells may be relocated and direcred to the desired locations, and cell unions and organoid-like structures can be enhanced. Acoustic cell assembly can be used to create simple cellular models like brain models [143] and disease models [144] (Figure 5a) for study. In addition, acoustic assembly of cells can be used to generate specialized organs using ultrasound and the appropriate materials. In addition to academic applications, there is potential for organ printing when the technology is mature enough for widespread use. 

When compared to optical imaging’s lessened capacity to penetrate live tissue, the acoustic field’s low attenuation represents a significant advantage. Microbubble targeting is a widely researched method. Scientists have constructed an internally inflated protein nanostructure that allows nondestructive imaging of target bacteria in mice at a resolution of less than 100 microns, as shown in Figure 5b [145].

Acoustically driven propulsion can penetrate in medium with low losses and thus provide a promising mode of transportation. Microbubbles, phase-change nanodroplets, and nanocarriers are the three methods for transporting medicine to the desired location within the body. Upon ultrasound exposure, microbubbles play a vital role in efforts to transport and release a chemical or nanoparticle payload. Typically, lipids and encapsulated perfluorocarbon gases are used to build a shell for drug delivery, allowing the payload to be dissolved in lipids and incorporated into the shell.

Nanodroplet is developed to address the deficiency of microbubbles as alternative carriers [147]. Since nanodroplet is vesicles containing a core of phase-changeable material, it can be decorated with functional drugs [148]. It is comparable in size to endothelial cells and increases drug uptake by enhancing extravasation into tumor tissues.

Nanocarriers present a third option for the ultrasound-triggered release of payloads, including various inorganic and organic particles with diameters as small as several hundred nanometers, allowing them to reach microscopic locations (Figure 5c) [146]. 

### 2.3. Chemical and Light-Induced Mechanism

The magnetic and acoustic field driven methods described above are used to manipulate micro/nanoparticles by an external field, and these systems rely on an external field to drive the particles, which are inert. Instead, scientists proposed employing chemical mechanisms to propel the particle, effectively turning it into microscopic motors that propels itself without the need for external assistance. However, since the microscale particles need to overcome strong viscous forces, the mechanical aspects of its properties should still be investigated. Normally, the drag force (F_η_) on a spherical particle with diameter d_s_ in a human-like environment is evaluated by Stokes law [30]:(9)$$F_{\eta }=3{\pi \mu d}_{s}\nu$$
where μ represents the coefficient of viscosity or the dynamic (or absolute) viscosity of the fluid, and v is the propulsion speed. As can be seen, there is a great deal of complicated work to be done in order to accomplish exact control of chemically propelled particles and functions like targeted drug delivery. For chemically catalyzed micro/nano motors, the particle itself acts as a catalyst to decompose water and produces substances such as oxygen or form chemical gradients to propel particles forward. For the purpose of controlling the occurrence and termination of the reaction, light is introduced to serve as a tool of a switch. The manipulation of the micromotors is mainly focused on particle material selection and strategy design, so this section will concentrate on them.

The aforementioned driving techniques are for single particles, yet in nature, creatures act in groups at the microscopic scales. Researchers find that this fantastic idea of collective behavior of microorganisms can apply to the microrobots to aggregate them together to complete the tasks. A variety of intriguing strategies for particle population management have been developed, which are listed detailed below. This part mainly focuses on the fundamental chemical and light-driven mechanisms and their biomedical applications in recent years.

#### 2.3.1. Basic Propulsion Mechanisms

The asymmetric structure could account for the behavior of chemically propelled single particle. Therein, the Janus particle, sourcing from the two-faced Roman God Janus, can be self-propelled because of this non-equilibrium property. Although it has different shapes like a rod, wire, disk, and commonly sphere, they all share similar functions. Several types of researches have been done to construct the Janus mode, among which spontaneous symmetry breaking is an effective measure. This section outlines three types of chemical and light-mediated driving methods, which are bubble propulsion, self-diffusiophoresis and self-electrophoresis.

A facile propulsion mechanism is bubble propulsion, as the micro/nanobubble can be generated and ejected from the surface of the motors to push itself. A successful landmark application fabricated by Wilson et al. stimulated the ambition to research in the manufacture of bubble propulsion particles [149]. Moreover, the special structure of this particle is just like a miniature monopropellant rocket with platinumloaded, as shown in Figure 6a. Pourrahimi et al. proved that the ZnO-Pt Janus motor could efficiently accelerate the fundamental ZnO particle [52]. Zhang et al. developed a jellyfish-like micro motor which is powered by a peroxidase-modified DNA structure. This design strategy exhibits incredible propulsion speed, as shown in Figure 6b [53]. Light controlled bubble propelled microrobots have been widely researched due to its manageability. Mou et al. found that the bubble will created from the inner surface of the TiO_2_ tubular particle under UV light [150]. In recent years, researchers found that visible light could be introduced to take the place of UV irradiation, greatly accelerated the advancement in bubble propulsion constructions [151]. 

Self-diffusiophoresis is a mechanism that produces chemical species from the hemisphere of the Janus particle continuously to generate a steady chemical gradient, which leads to the movement of the motor towards the inert side [30]. For example, in hydrogen peroxide solutions, the generated chemical gradient is able to drive the particles due the following reaction:(10)$$2H_{2}O_{2}\to 2H_{2}O+O_{2}$$

Wilson et al. explored a platinum-loaded stomatocytes that can exist in both self-diffusion swimming and bubble propulsion systems in the preceding bubble propulsion section [149]. The essential difference between these two methods is whether the gas discharged from one end of the nanoparticle generates a self-propelling bubble. It is a self-diffusive swimming mechanism if only a chemical concentration gradient is formed. Recently, Yu et al. manipulated Janus-like dimer motors that assemble spontaneously [152]. Figure 6c briefly shows the assembly process under UV light, which is assembled through diffusiophoretic interactions between the chemically-active symmetric colloid and the passive one. A photochemically propelled poly-(methyl methacrylate) (PMMA)-AgCl Janus micromotor was developed by Zhou et al. [54]. They proposed a model system to better understand the self-diffusiophoresis mechanism. In simple terms, it is the photodecomposition of AgCl and the movement away from the AgCl side, which is shown in Figure 6d. Mou’s team exploited a new construction using CO_2_ to serve as fuel to propel the particles [55]. This method significantly promoted application development in vivo since CO_2_ is a biocompatible gas for humans. To make the self-propelled biocompatible motors more applicable, Singh et al. discovered the photogravitaxis of a photochemically active particle which can be manipulated by light in 3-dimention [156], creating more possibilities for follow-up research.

Inspired by the mechanism of enzyme catalysis in organisms, scientists proposed a new approach of immobilizing enzymes on the particle to generate the chemical gradient. Ma et al. designed a bio-catalytic Janus mesoporous silica nanomotor, which is the first successful work to fabricate the under 100 nm enzyme motor [14]. Followed by, these tiny micromotors, which offer a lot of potential in biological applications, have been thoroughly investigated. Beside of catalase, researchers investigated various types of enzymes such as urease and lipase [157,158]. Choi et al. examined an enzyme-powered polymer motor that has been successfully applied to a mouse bladder [57].

The contrasts and similarities between the two processes of self-diffusiophoresis and self-electrophoresis have been widely debated. Neutral self-diffusiophoresis and ionic self-diffusiophoresis are the two basic forms of these mechanisms. Chemical reactions on the particle surface provide a concentration gradient, which in turn causes a tangential pressure gradient, thus drives fluid flow to propel the particles forward in both modes. Self-electrophoresis differs from self-diffusiophoresis in that it produces proton flow, which is not present in the diffusiophoresis mechanism.

The self-electrophoretic process is considered to play a crucial role in asymmetric Janus particles wrapped with one photoactive side and a novel metal side, such as Pt and Au in H_2_O_2_ solution [159]. The primary reactions on the Janus particle are:(11)$$H_{2}O_{2}+2e^{-}+2H^{+}⇄2H_{2}O\left(\mathrm{Reduction},\;\mathrm{Au}\right)$$
(12)$$H_{2}O_{2}⇄2e^{-}+2H^{+}+O_{2}\left(\mathrm{Oxidation},\;\mathrm{Pt}\right)$$

From these equations, it can be seen that a flow of rapid protons from the Pt end to the Au end is continuously generated, which result in the waterflow to push the particles forward. At the early stage of the research in this field, Paxton et al. explored a rod-shaped particle with these two segments [160]. The asymmetrical structure of the particle TiO_2_-Pt Janus is also an effective strategy due to its perfect heterostructures and activity of light catalyst [58]. It will undergo a photocatalytic redox reaction with the fuel of H_2_O or H_2_O_2_. The separation of the photogenerated charge pairs in heterojunctions of motors will promote the occurrence of opposite charges on the two sides of the aggregates and the different segments of the motor both serve as electrodes. Brooks et al. introduced a mathematical model to explain the phenomenon. To visualize this mechanism, they designed the particles to take the shape of micro-gears and showed that the Pt catalytic micromotors would spontaneously rotate at a certain angle and speed, and the angular velocity is changed with the concentration of salt in the solution [161]. Through programming and controlling the zeta potential of the Janus nanotree-particle, Dai et al. fabricated a novel method to change the phototaxis of the swimmers and propelled them forward [153]. The tree branch-like structure of the motor is demonstrated in Figure 6e.

Light is a useful tool to control the motion of particles in the propulsion of self-electrophoresis. Dong et al. precisely manipulate TiO_2_-Au Janus particles under fuel-free, low light energy conditions [59]. With a thin film of Au on the TiO_2_ particle, as shown in Figure 6f, the particle has good biocompatibility for practical application. However, the cost of the materials like Pt and Au is too high to afford in most cases. To reduce production costs, Wang et al. put forward a program that uses Fe to take the place of the expensive metal [60]. It holds great potential in future applications.

#### 2.3.2. Swarming Behavior

Swarming behavior has attracted the attention of researchers for decades. As discussed before, chemotaxis is a sticking point of self-propulsion and assembly. The particles tend to move directionally of the chemical gradient and would have an influence on the other by releasing substances, dynamic behaviors, autonomously responding to the environment, etc. Ibele et al. proposed an idea of the method of intelligent assembly of AgCl particles, which provided a research direction of the strategies about how to assemble the nanoparticles by themselves [162]. Another noteworthy mechanism is self-aggregation which is inspired by the bacterial flora. This phase of the cluster is dynamic, and the cluster size depends on the propulsion speed of the particles as well as other parameters, which is confirmed by the work of artificial self-propelled colloids done by Theurkauff [163]. Since then, many outstanding achievements in microrobotic swarming have been made to greatly promote the development of in vivo application [164]. This part mainly discusses the swarming mode and the respective advantages of Janus particles.

For the non-equilibrium particles, the basic installing principle is the chemically active particles’ attraction to the passive one, then the aggregates grow up to become a multi-layer shell. In turn, the passive particles can influence the behavior of the active one, forming a complex combinatorial structure. Some of the significant discoveries about the approaches for controlled and stable binding of particles are shown below.

The swarming behavior appears due to the interactions among the single particles and cooperatively finishing the tasks. To gather the micromotors, Singh et al. used chemical-activated TiO_2_-SiO_2_ Janus particles which can self-propelled under UV light to condense the distributed passive collides controlled by UV light [154]. Figure 6g illustrates the growing process of the first shell of one aggregate, then the tiny shell grows up by attracting more passive (white) collides. However, this kind of walnut-like robots exhibit some defects in the speed and controllability of movement. Peng et al. compared the difference of swarming behaviors between walnut-like and cubic-shaped hematite microparticles, and they found that by being able to combine into three different chain configurations under UV light, the cubic microrobots not only move faster but can also adapt to more demanding scenarios [165]. Ketzetzi et al. built a simple train model to more thoroughly investigate the chain robot swarm’s collective behavior and discovered several intriguing phenomena [166]. Each particle is automatically spaced properly, and it is possible to speed them by increasing their number, altering the curvature of their motion trajectories, etc.

As mentioned in Section 2.3.1, the rotation property (speed and direction) of plate-like Pt particles based on the self-electrophoresis method is due to their shape and the extent of non-equilibrium [161]. This work mainly focuses on single-particle rotation, and Aubret and his team’s research is about self-spinning swarming particles [155]. They showed that the light-driven self-powered microgears can hierarchical assembly by themselves, and the installing process is illustrated in Figure 6h.

In addition, we need to consider the impact of the environment on the movement of the microrobots. For microswimmers moving through liquid, the Marangoni effect, which can dominate the particle’s dynamics, is a notable corollary [167]. A single active colloidal particle forming a chemical gradient encounters an effective force of hydrodynamic origin near a fluid-fluid interface. This force is caused by fluid flow driven by Marangoni stresses, and the velocity carried by a flow of an active particle at a distance L from the interface is:(13)$$\nu \left(L\right)\approx u\left(x_{0}\right)\approx -e_{z}\frac{{\mathrm{Qb}}_{0}}{64{\pi D}_{+}\eta_{+}L}$$

Here $\nu \left(L\right)$ denotes the translation velocity of the particle, $u\left(x_{0}\right)$ is the Marangoni flow, Q denotes the total rate of product release by the particle, η_+_ denotes the average of the viscosities of the two fluids, D_+_ denotes the average of the diffusion constants in the two fluids of the product of the chemical reaction, b_0_ is the coefficient of tensioactivity relating, in linear approximation, the changes in the surface tension to the changes in the number density of the reaction product at the interface. On the basis of this finding, a sessile drop containing titania powder particles have been studied in depth by Singh et al. [168]. The change in chemical composition can alter the surface tension of the drop, which induced relevant Marangoni stresses. The results of this study demonstrate a transition to collective motion, resulting to self-organized flow patterns.

Research in recent years has focused more on creating sophisticated and programmable robotic swarming systems. The interactions inside the motor are vulnerable to Brownian motion, which is a non-negligible flaw present in most assemblies. Because the isotropic particle light orientation is stable and may offset the effects of Brownian motion, Che et al. then created a light-programmed system for motor interactions, which offers up a slew of new possibilities for future research [169]. In addition to the manipulation by means of light, a kind of fluid dynamics based micromotors for communication has also been studied [170]. A single micromotor can generate a strong, steady flow of polymeric electrophoresis to trigger neighboring motors for mutual or unidirectional, controlled or spontaneous communication. These works offer up a slew of new possibilities for future research.

#### 2.3.3. Biomedical Applications

Chemical and light-induced driven particles show great potential in environmental remediation and biomedical engineering due to their high controllability and biocompatibility [171]. The direction of micro/nano motors in biological applications has therefore been the subject of intense research in recent years. However, when applied to human beings, there are several concerns that must be taken seriously. For example, the toxic fuel caused by the fuel of hydrogen peroxide and the incompatible, not degradable materials of the nanomotors might lead to immune response or inflammation [6]. Significant solutions are explored in recent years. Substances inside human body appeared as biocompatible fuel, such as enzymes. Meanwhile, fuel-free driving can be achieved by adding an external light field, and this method has also been widely used to reach the goal of biocompatibility, biodegradability, and circulation stability. This section carefully discusses the contributions of micro-nanorobots to the medical and health fields in recent years, and adapts to the desires for pleasant and painless treatment and inspection of health disorders. 

Low therapeutic efficacy and significant adverse side effects in drug delivery have troubled the health treatment system for decades [172]. Drug-delivery nanovehicles sending medicine to the target position has been constructed to address this problem and show fascinating results. Hortelão et al. brought forward urease-powered mesoporous silica shell nanomotors to load and deliver the anticancer drug doxorubicin [173]. The experiment exhibits favorable propulsion competence in ionic media (PBS buffer), and Figure 7a illustrates the fabrication process and some features of this urease nanorobot. Modern surgery improvement is a transformation from large equipment to micro-nano scale invasive therapy thus using microelectromechanical systems technology to fabricate the micromotors attracts a lot of attention [174]. 

Interacting with the biosystems and regulating cell activities is a brand new field developed by Chen et al. [175]. They find a UV irradiation (365 nm) TiO_2_-Au nanowire (NW) motors capable of high precision and accurate activation. This discovery inspired us that nanomotors can provide the driving force and generate bio-signals to regulate the activity of neuron cells. Figure 7b displays the assembly procedure of the TiO_2_-Au NW motors and cell activation process under UV light.

A susceptible, rapid, low-cost and straightforward detection of target biomarkers and disease surveillance such as immunoassay method is based on the detection of motion properties of nanomotors [35,36]. Measuring the surroundings of the particle, such as the pH of the solution environment, is one indirect technique to track particle mobility. Patino et al. discovered a DNA nanotechnology and urease-powered micromotor with pH sensing function [176]. With regards to this, a fluorescence resonance energy transfer (FRET)-labeled triplex DNA nanoswitch was embedded on the surface of mesoporous silica-based micromotors. Figure 7c exhibits the negative correlation between propulsion speed and the sensing pH.

For practical in vivo application, Gao et al. accomplished the pioneered work in a mouse’s stomach using a Zn-based micromotor. This particle effectively releases the cargo and auto-degradation in gastric acid, which is no toxic to living creatures [178]. The swarming enzymatic nanomotors also are applied to the bladder, which constitute a crucial milestone in practical medical imaging tracking area [179]. According to de Ávila et al., they accomplished the first in vivo task to use the therapeutic micromotors to treat stomach bacterial infection in a mouse model (Figure 7d) [177]. In the acidic medium of the stomach, the drug-loaded magnesium-based micromotor is capable of autonomous propulsion with precise positioning and drug delivery. Despite the promising outcomes of the above-mentioned application scenarios and approaches, chemically driven micro/nanomotors are still in their early stage of development, and more study is needed to improve and enhance them. The promising findings obtained in the above-mentioned application scenarios have the potential to be applied to future human applications.

### 2.4. Hybrid Propelling Mechanism

During the exploration of micro/nanorobots, researchers have discovered that more precise modulation in a single driving mode is limited. However, the hybrid driving modes, under multiple power sources, can provide the guarantee of controlled motion for the microrobot’s self-propulsion, which can better satisfy the requirements in practical application conditions. Hybrid field driven micro/nanorobots are usually driven by a combination of two or more distinct external fields like magnetic, acoustic, optical, and chemical to be controlled synergistically. This hybrid method can produce a more efficient and precise carrier transport process and better biocompatibility to adapt to a more complex environment. In this section, hybrid drive systems for magnetic-acoustic, acoustic-chemical, and magnetic-chemical/optical propulsion will be discussed.

#### 2.4.1. Magnetic-Acoustic Fields

While chemical propulsion has attracted tremendous interest in driving catalytic motors, it typically exhibits autonomous self-propulsion powered by chemical fuels (e.g., hydrogen peroxide, acid/base fuels, etc.) [180,181]. However, many practical applications of micro/nanomotors require the elimination of external fuels, notably in vivo biomedical applications with high demands for biocompatibility. To address these needs, recent efforts have led to the development of biocompatible fuel-free propulsion mechanisms based on external stimuli. Improving the capability and sophistication of such fuel-free nanomotors is critical for designing advanced micro/nanosystems for various biomedical applications. The two main characteristics of acoustic and magnetic fields are biocompatible and contactless, contributing to effective and precise propulsion.

In the recent research reports, segmented gold-ruthenium nanorods with thin Ni segments at one end can be propelled by 4 MHz ultrasound. In addition, due to the interaction between nickel parts, monomers can be assembled into multimers driven by magnetic fields to produce different modes of motion [131]. Feng et al. designed a magnetic microrobot with ultrasonic levitation. Compared with the pure magnetic field controlled microrobot, the 3-D rotation of a single oocyte has a significant improvement in control accuracy and rotation speed [182]. Mohanty et al. designed CeFlowBot, a bubble-powered cephalopod-inspired untethered microrobot capable of locomotion in fluids [183]. The microrobot contains an array of six entrapped air bubbles in its body that vibrate to provide a directional flow via its inner channel. In addition, they synthesized magnetic layers within CeFlowBot, allowing it to be guided by uniform magnetic fields while propelled by acoustic fields. And they demonstrated the locomotion of this microrobot and their ability to grasp and release a payload under acoustic fields, as shown in Figure 8a. Li et al. designed a hybrid nanorobot based on a two-segment structure, with magnetic nanosprings (Ni-coated Pd) and concave nanorods (Ni-coated Au) forming the two ends, respectively [184]. The magnetic helical ends are used for magnetic propulsion, and the concave nanorods are used for acoustic propulsion. Here a phenomenon as shown in Figure 8b is observed, where the acoustic field generated by the pressure gradient can trigger the clustering behavior of hybrid nanomotors, while the collective clusters of nanorobots under the magnetic field disintegrate and are propelled into directional motion. The movement patterns and directions of these particles may be dynamically changed by varying the application of external fields. 

With the combination of acoustic and magnetic fields, the collective behavior of micro/nanoparticles can be applied to targeted drug delivery. A sequential micro/nano-robotic drug delivery process was realized by Jeong et al. [61]. The microrobot consists of a cylindrical neodymium magnet and two microtubes holing the gaseous bubble. Figure 8c depicts that after the microrobot completes drug transmission to the required location under the control of a magnetic field, the bubble carrying the drug completes drug release under the assisted control of an acoustic field. A microrobot mimics the collective behavior of neutrophils, which is capable of particle self-assembly in a magnetic field and can be manipulated with acoustic fields in the vascular system, thus achieving a rolling-type motion along the boundaries [62]. Gao et al. proposed a strategy of an acoustically powered and magnetically navigated red blood cell-mimicking (RBCM) microrobots based on the photodynamic therapy (PDT) method. This RBCM microrobot can actively transport oxygen and photosensitizers to promote the efficiency of PDT [41]. 

#### 2.4.2. Acoustic-Chemical/Optical Fields

Due to the non-contact properties and compatibility of acoustic fields with biological systems, researchers have used acoustic fields to catalyze efficient and reversible clustering. Also, the acoustic fields could control the motion of motor separation as well as trigger the assembly of chemically powered nanomotors. The chemodynamic nanomotors migrate in the acoustic field due to acoustically generated pressure gradients that cause them to migrate towards the low-pressure region (nodes or antinodes). Ultimately, these motors are rapidly localized and aggregated in the low-pressure region. The new clustering process is speedy and reversible. This ability to assemble catalytic nanomotors and regulate their collective behavior offers considerable promise for the fields of cargo transportation, nanomechanics, and chemical sensing.

The acoustic microcannons, as a new tool for micro-scale tissue penetration, can achieve a high-speed motion [187]. In the research work where acoustic field and chemical drive modes were used, Xu et al. used an acoustic field to trigger the assembly of chemically powered nanomotors; the schematic diagram of the clustering behavior is shown in Figure 8d [67]. Also, they researched the efficient and reversible swarming and separation of nanomotors of different types. When the ultrasound is turned on, the collective behavior of chemically powered Au-Pt nanomotors can be seen under the microscope. The acoustic radiation forces theory demonstrates that the migration of catalytic nanomotors results from the low-pressure regions. The nanomotors generate collective behavior based on the acoustically generated pressure gradients. Due to autonomous catalytic propulsion, the cluster of nanomotors rapidly disperses when the ultrasonic field is switched off. The controlled motion of the entire nanomotor assembly is achieved by modulating the variables of the acoustic field (e.g., frequency or voltage). Ren et al. designed a device integrating chemical fuel and acoustic force, which can both positive and negative rheotaxis of synthetic bimetallic micromotors [66]. As can be seen from Figure 8e, the directional collective motion of bimetallic Rh-Au micromotors in the chemical and acoustic fields was different, respectively. Under the acoustic and chemical fields, the bimetallic micromotors can mimic the rheotaxis behaviors of their natural counterparts, which represents a significant step in biomedicine. 

The microrobots can also be propelled by acoustic and optical fields in synergy. The acoustic field propels the micromotor made of Au and TiO_2_. With the presence of hydrogen peroxide and UV light, the chemical reactions on the Au and TiO_2 _surface contribute to the motion of the microrobots [64]. To mimic biological systems’ aggregation/separation behavior, a light-acoustic combined strategy is designed to propel the nanomotors [65]. The collective behavior of the artificial nanorobots is driven by the acoustic field, while the collective outward firework-like motion of the nanorobots is observed when the cluster is induced by light irradiation, which is accomplished due to the change of acoustic streaming by optical force; this variation can be seen in Figure 8f. The cluster’s diffusion velocity and region can be controlled by alternating the light intensity, acoustic excitation voltage, and irradiation direction. Also, the diffusion shape of the “firework” can be governed by the irradiation position and angle. This novel strategy applies to metallic, metalloid, and polymers, representing a promising future for chemical sensing, cargo transport, and other biological applications. Xu et al. designed an assay platform that exhibited enhanced Raman signals by integrating acoustic aggregation of modified Au nanorobots, facilitating ultra-trace rapid biomolecule detection in microliter solution [188]. This strategy with ultrasound aggregation-induced enrichment (UAIE)-based surface-enhanced Raman scattering (SERS) demonstrates low sample usage, fast response, and low detection limit, offering a promising future for accurate biomarker identification and precise diagnosis of diseases. Wang et al. reported a nanorobot consisting of a needle-like gallium core wrapped with an anticancer drug as well as an outer leukocyte membrane shell used as a functionalized modification [63]. The asymmetric needle-like structure and the high density of gallium-based liquid metal are responsible for the strong propulsion. The speed and direction of motion of the nanorobot can be regulated by the voltage and frequency of ultrasound. In addition, the coating of leukocytes avoids biological contamination during the propulsion process, while being able to identify cancer cells and improve the efficiency of photothermal and chemical therapy.

Photoacoustic computed tomography (PACT) is an imaging technique that uses infrared laser pulses to conduct more in-depth real-time visualization as well as accurate in vivo monitoring in microsurgery. Wu et al. designed a PACT-guided microrobotic system (PAMR) [185]. With the help of PACT images, they can locate tumors in the digestive tract and track the location of microrobots, which accomplished controlled propulsion and extended in vivo cargo retention; a process is shown thoroughly in Figure 8g. This magnesium-based microrobot encapsulated in a microcapsule enables the active migration of the microrobot in the gut to be observed in real time by photoacoustic imaging. When the microrobotic capsules overcome the gastric acid barrier to reach the patient area of the intestine in vivo, exogenous near-infrared light can penetrate deep into the tissue and trigger the rupture of the capsules to release the microrobots. The released microrobots are exposed to digestive fluid, and the magnesium sphere reacts with the digestive fluid chemically to produce tiny bubbles that propel the spheres and ultimately achieve retention and continuous drug delivery in the patient area.

#### 2.4.3. Magneto-Chemical/Optical Fields

Combined chemical or optical and magnetic fields are also propulsion methods that have received much attention. This method can solve the problem of fuel depletion during a prolonged catalytic operation and control the motion direction of the micro/nanoparticle under the magnetic fields.

The swimming microrobot optical nanoscopy utilizes chemical locomotion and magnetic guidance for nondestructive scanning and imaging over large areas [189]. The biocompatible microrobots, iMushbots, composed of mesoporous mushroom (Agaricus bisporus) fragments coated with magnetite nanoparticles (FeONPs), could be used to deliver anticancer drugs under peroxidase induction as well as magnetic field guidance [190]. A hybrid nanorobot consisting of flexible multi-segment Pt-Au-Ag-Ni was designed by Gao et al. [191]. By switching between catalytic and magnetic propulsion, the conflict of slower movement due to fuel depletion and high-ionic-strength media is mitigated. Jiang et al. proposed a dual-driven biomimetic microrobot by depositing Au and Ni nanoparticles on the surface of the microrobot enabling responses to external optical and magnetic stimulus. An infrared laser beam and electromagnetic gradient field are utilized, respectively [192]. In detail, macroscopic manipulation is performed by controlling the electromagnetic gradient field, while the infrared laser beam projecting to the surface of the microrobot finely orients it to a predefined location. 

Villa et al. designed the superparamagnetic/catalytic microrobots (PM/Pt microrobots) made of superparamagnetic polymer particles [68]. It consists of interior iron oxide as well as a tosylated surface exterior, which is partially covered by a catalytic platinum (Pt) layer. The function of the microrobots can be divided into three parts, bind molecular and biological materials by tosyl group-rich polymer layer; catalyze the decomposition of hydrogen peroxide to power the propulsion of the microrobots by Pt layer; control the direction of the microrobots under the magnetic field by the magnetic part. As shown in Figure 8h, the superparamagnetic cores of these microrobots enable them to assemble in long chains to move, capture and transport breast cancer cells under an external magnetic field, demonstrating their application in cell transportation of antitumor drugs. As shown in Figure 8i, Bozuyuk et al. designed a chitosan-based double-helical microswimmer propelled by the magnetic field, which releases doxorubicin, a chemotherapeutic drug, with the help of external light stimulus [70]. The team revealed a novel strategy that combines the motion of the microswimmer under the magnetic fields with a light-triggered drug release mechanism that reduces the side effects of the entire microsystem in minimally invasive therapy. As shown in Figure 8j, Yuan et al. proposed the two-dimensional nanomaterial-coated micromotors that integrate three engines controlled by three different stimuli (chemical fuels, light, and magnetic fields) [186]. As for the engines, bubble (catalytic)-propulsion is triggered by Pt or MnO_2_ nanoparticles; magnetic mode handles are introduced by Fe_2_O_3_ nanoparticles, and light mode engines are controlled by quantum dots. This synergetic effect between the three engines brings increased speed and an efficient propulsion rate.

## 3. Discussion and Conclusions

Micro/nanorobots are developed based on the propulsion mechanism of biological microorganism, and they are constantly refined and improved to meet the needs of biomedical applications. This review highlights the four basic propulsion mechanisms of micro/nanoparticles that have been demonstrated in the last decade, as well as their individual propulsion properties and application situations.

In Section 2.1, we have discussed the actuation of micro/nanorobots driven by magnetic fields. For the control of a single magnetically active microrobot, the driving methods can be classified into three categories, which are corkscrew-like motion, traveling-wave locomotion and surface assisted motion. For the control of microswarm, the method of triggering the collective behavior can also be classified into three categories, which are magnetic dipolar interactions, hydrodynamic interactions and weak interactions. The approach of magnetic field control exhibits significant advantages for managing the population of microrobots, since magnetic manipulation is a robust method for remote control [42,43]. Moreover, the utilization of magnetic fields to impel the microrobot is fuel-free, which eliminates the impact of toxic chemical products harming the cells and tissues during the biological application [89]. Another strength is that using micro/nanorobots to transport therapeutic substances under magnetic fields can enhance the performance of hyperthermia therapy [113], cancer therapy [114], ophthalmic treatment [116,117], etc.

In Section 2.2, we have compared two major particles’ propelling categories applied in acoustic field, the acoustic streaming and the acoustic radiation force. Acoustic field has shown some practical advantages, the first of which is safety considering the safe frequency range that the acoustic field covers. The high precision and low losses while travelling through complex bio-median indicate another advantage of using acoustic fields, which show great potential for biomedical applications. 

In Section 2.3, we have introduced chemically/optically driven micro/nanorobots, which have mostly been classified into three categories: bubble propulsion, self-diffusiophoresis and self-electrophoresis. These propulsion systems often employ an asymmetric structure known as the Janus particle. The development of Janus particles is noteworthy because of their simple fabrication and amazing physical properties unique to broken symmetry [193]. Scientists have devised a number of novel approaches to better fit the use in the human body environment, such as employing carbon dioxide as a fuel, gold as a reactant material, etc. [55]. In vivo tests in mice bladders have also yielded promising results [158,173].

In Section 2.4, we discuss the hybrid driven systems for magnetic-acoustic, acoustic-chemical, and magnetic-chemical/optical propulsion. Hybrid micro/nanomotors utilize various power sources and can increase the efficiency of micro/nanorobots in constantly changing conditions. As a result, this propulsion mechanism is more effective because it can enhance drug-carrying capacity, motility, and signal feedback for medical imaging. As the field of micro/nanorobotics research becomes more mature and new achievements continue to emerge, hybrid micro/nanorobots will benefit us in biomedical applications [61,68,69,185]. 

Micro/nanorobots are widely developed for precise drug delivery and targeted treatment of diseases, as proven by some of the advances that have been made in bio-nanomedicine. Here, we discussed typical driving methods of micro/nanorobots and their operating mechanisms, and highlighted the contribution of using individual and swarming micro/nanorobots as novel and effective diagnostic and therapeutic tools in the biomedical area. By reviewing representative research works, it serves to propose suggestions for achieving more precise control, furthering its application in realistic environments inside living organisms, and enhancing interoperability among clustered micro/nanorobots. The field of nanotechnology has made remarkable progress in the last decade, but research on micro/nanorobots is still in its infancy. The more experimental study is needed to determine if these small robots can work reliably and constantly in the body for an extended period of time, whether human immunological rejection would hurt the body, and how diverse human surroundings will affect the life of the microrobots. Advances in micro/nanorobots in biomedicine allow us to envision a future in which tiny robot physicians can go clinical to assist human surgery and play a greater role in health care. 

## Figures and Tables

**Figure 1 micromachines-13-01473-f001:**
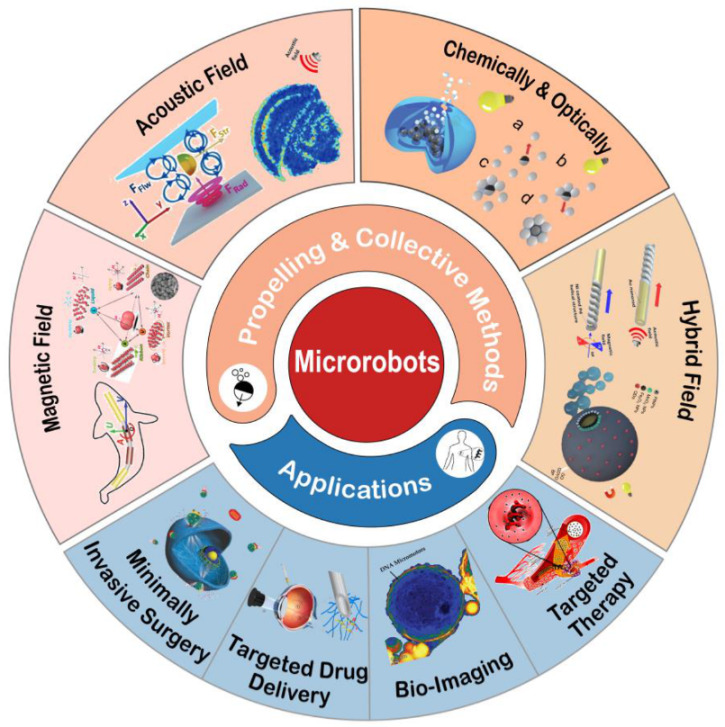
Schematic illustration of the key features and application scenarios of micro/nanorobots.

**Figure 2 micromachines-13-01473-f002:**
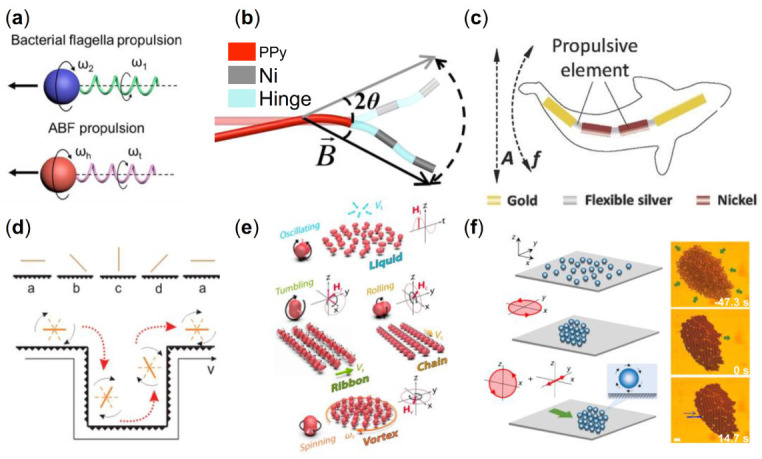
Schematic illustration of magnetic propulsion mechanism. (**a**) Schematic diagram bacterial flagellum and ABF propulsion mechanism, where ω_1_ and ω_2_ have different rotation direction while ω_h_ and ω_t_ have the same rotation direction. [89] (Reproduced with permission from Zhou et al., Chemical Reviews; published by American Chemical Society, 2021); (**b**) Schematic diagram of 3-link nanoswimmer with undulation motion under a magnetic oscillating field (2θ = angular sweep of the magnetic field) [96] (Reproduced with permission from Bradley et al., Advanced Materials; published by Wiley, 2012); (**c**) Schematic diagram of an artificial nanofish consisting of gold, silver, nickel, silver, nickel, silver, and gold segments [72] (Reproduced with permission from Li et al., Small; published by Wiley, 2016); (**d**) Schematic diagram of nanowire surface motion robots [97] (Reproduced with permission from Wang et al., Small; published by Wiley, 2016); (**e**) Four collective formations—liquid, chain, vortex, and ribbon—were programmatically triggered by alternating magnetic fields [25] (Reproduced with permission from Xie et al., Science Robotics; published by Wiley, 2019); (**f**) Schematic diagram of magnetically propelling colloidal carpet actuated by rotating magnetic fields and images of carpet assembled by a magnetic rotating field under an optical microscope [98] (Reproduced with permission from Helena Massana-Cid et al., Nature Communications; published by Springer Nature, 2019).

**Figure 3 micromachines-13-01473-f003:**
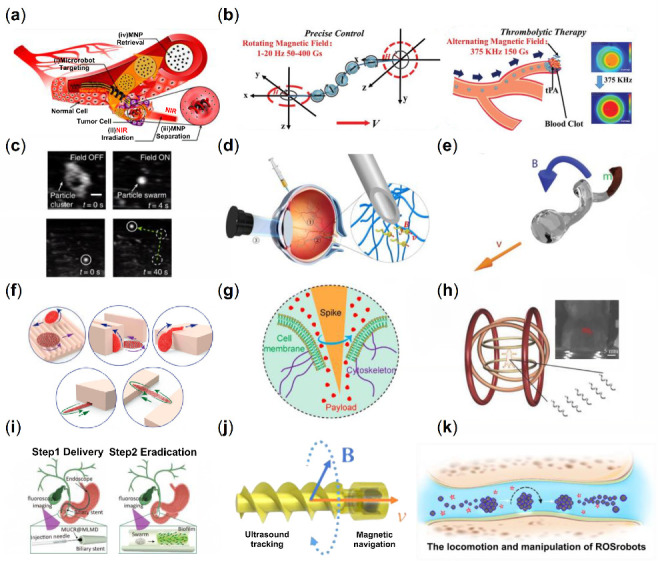
Schematic illustration of biomedical application based on magnetic fields. (**a**) Schematic diagram of the targeted cancer therapy process of helical microrobots [39] *(*Reproduced with permission from Lee et al., Applied Material and Interfaces; published by American Chemical Society, 2021); (**b**) Schematic diagram of a BMM with magnetic separation adjustment for targeted thrombolysis [114] *(*Reproduced with permission from He et al., Advanced Materials; published by Wiley, 2020); (**c**) Schematic diagram of illustrating the generation of a medium-induced swarm in the vitreous humor and the trajectories of the navigated locomotion of the swarm [116] *(*Reproduced with permission from Yu et al., Nature Communications; published by Springer Nature, 2019); (**d**) Schematic diagram of the slippery micropropellers’ three-step targeted delivery process [117] *(*Reproduced with permission from Wu et al., Science Advances; published by American Association for the Advancement of Science, 2018); (**e**) Schematic diagram of navigation, where m is a high magnetic moment and B is a small external rotating magnetic field [118] *(*Reproduced with permission from Fischer et al., Advanced Materials; published by Wiley, 2020); (**f**) Schematic diagram of navigation movements adapted to multi-terrain conditions [121] *(*Reproduced with permission from Sun et al., Advanced Functional Materials; published by Wiley, 2022); (**g**) Schematic of single-cell perforation and payload delivery in dual-action [122] *(*Reproduced with permission from Xie et al., Small; published by Wiley, 2020); (**h**) Image of a 4-week old Balb-C mouse under anesthesia inside the magnetic coils. The red spots represent the fluorescent signal of the injected f-ABFs [19] *(*Reproduced with permission from Bradley et al., Advanced Materials; published by Wiley, 2015); (**i**) Schematic diagram of two steps (delivery and eradication) using MUCRs swarm [123] (Reproduced with permission from Sun et al., Advanced Materials; published by Wiley, 2022); (**j**) Schematic diagram of propulsion mechanism about ultrasound tracking and magnetic navigation [124] (Reproduced with permission from Wang et al., Nano; published by American Chemical Society, 2022); (**k**) Schematic diagram of the mechanism of ROS-scavenging nano-robots (ROSrobots) for alleviating osteoarthritis. The removal of ROS from the synovium could inhibit the proliferation of M1 macrophages and increase the number of M2 macrophages to treat osteoarthritis [125] (Reproduced with permission from Zhao et al., Advanced Intelligent Systems; published by Wiley, 2022).

**Figure 4 micromachines-13-01473-f004:**
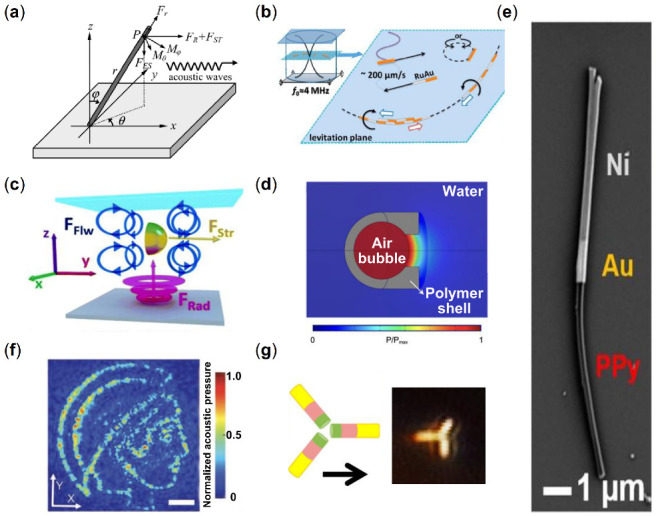
Schematic diagram of acoustic field drive method. (**a**) A nanotube connected to the surface of the substrate at one end, with pressures and torques applied to it by the incoming sonic wave and electrostatic contact [127] (Reproduced with permission from Lim et al., The Journal of Chemical Physics; published by AIP Publishing, 2007); (**b**) Schematic diagram of the trajectory and velocity of the nanorods [46] (Reproduced with permission from Wang et al., ACS Nano; published by American Chemical Society, 2012); (**c**) Nanoshells’ motion under the effect of ambient fluid flow(F_Flw_)and acoustic radiation forces (F_Rad_) [47] (Reproduced with permission from Soto et al., Nanoscale; published by Royal Society of Chemistry, 2016); (**d**) Simulated acoustic pressure diagram of a microrobot in a fluid medium [50] (Reproduced with permission from Aghakhani et al., Proceedings of the National Academy of Sciences; published by National Academy of Sciences, 2020); (**e**) Image of a nanoswimmer under scanning electron microscope [51] (Reproduced with permission from Ahmed et al., Nano letters; published by American Chemical Society, 2016); (**f**) The sound pressure field measured in the image plane [130] (Reproduced with permission from Melde et al., Advanced Materials; published by Wiley, 2020); (**g**) Shape of acoustic nanoparticle distribution Octahedral hexamer-shaped nanorods formed by the ultrasonic field [131] (Reproduced with permission from Ahmed et al., ACS Nano; published by American Chemical Society, 2014).

**Figure 5 micromachines-13-01473-f005:**
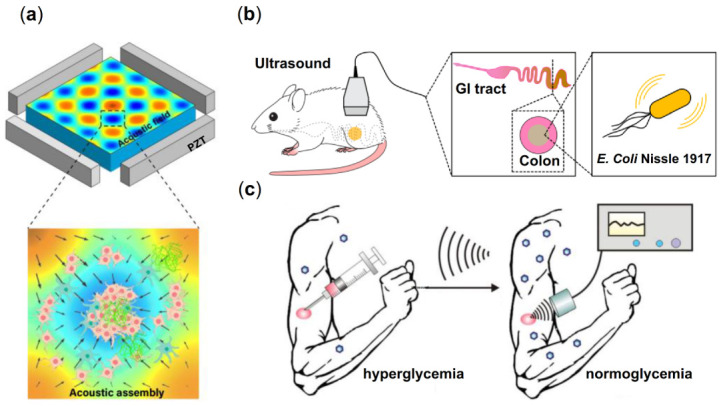
Schematic illustration of the applications of acoustic micro/nanoparticles (**a**) Acoustic assembly process of three-dimensional neurospheres [144] (Reproduced with permission from Cai et al., Analyst; published by Royal Society of Chemistry, 2020); (**b**) Experimental diagram of gastrointestinal tract imaging [145] (Reproduced with permission from Bourdeau et al., Nature; published by Nature Portfolio, 2018); (**c**) Schematic representation of long-term drug delivery after subcutaneous injection of nanocarriers [146] (Reproduced with permission from Di et al., Advanced Healthcare Materials; published by Wiley, 2014).

**Figure 6 micromachines-13-01473-f006:**
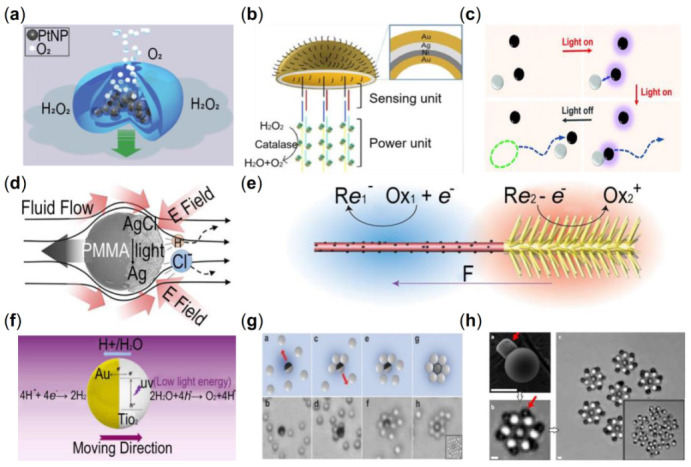
Schematic illustration of chemical with light-induced propulsion and swarming methods. (**a**) A profile of the platinum-loaded stomatocyte [149] (Reproduced with permission from Wilson et al., Nature Chemistry; published by Nature Portfolio, 2012); (**b**) The basic structure of jellyfish-like nanomotor produced by Zhang et al. [53] (Reproduced with permission from Zhang et al., ACS Applied Material and Interfaces; published by American Chemical Society, 2019); (**c**) Process of the chemically active and passive colloids under UV light [152] (Reproduced with permission from Yu et al., Chemical Communications; published by Royal Society of Chemistry, 2018); (**d**) A model system of propulsion of PMMA particle to better understand self-diffusiophoresis mechanism [54] (Reproduced with permission from Zhou et al., Langmuir; published by American Chemical Society, 2018); (**e**) Illustration of the structure of the Janus nanotree-particle [153] (Reproduced with permission from Dai et al., Nature Nanotechnology; published by Nature Portfolio, 2016); (**f**) TiO_2_–Au Janus micromotor propelled under UV light by catalyzing water [59] (Reproduced with permission from Dong et al., ACS Nano; published by American Chemical Society, 2016); (**g**) Growth of the first shell of the non-equilibrium assembly of the swarming particle [154] (Reproduced with permission from Singh et al., Advanced Materials; published by Wiley, 2017) (**h**) The rapid assembly process of self-spinning swarming particles is triggered by a focused laser beam [155] (Reproduced with permission from Aubret et al., Nature Physics; published by Nature Portfolio, 2018).

**Figure 7 micromachines-13-01473-f007:**
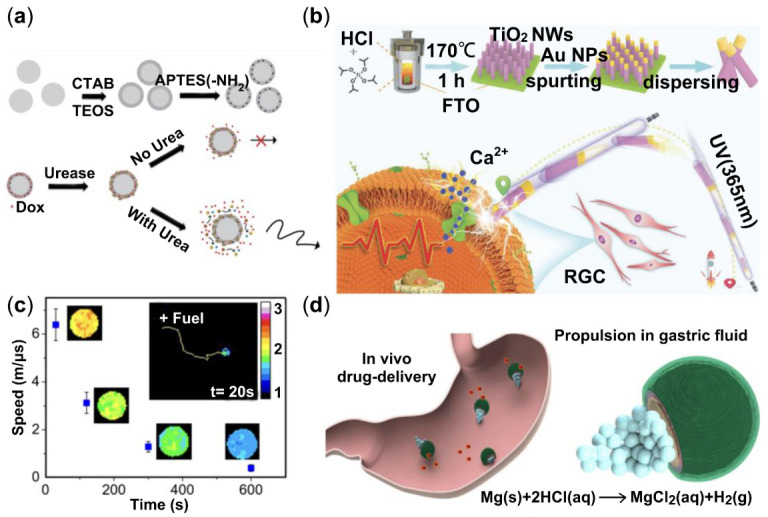
Schematic illustration of chemical-induced propulsion in biomedical application. (**a**) Schematic diagram of the fabrication process and some features of urease-powered nanorobots [173] (Reproduced with permission from Hortelão et al., Advanced Functional Materials; published by Wiley, 2018); (**b**) Method of the assemblage of TiO_2_-Au NW and the photoelectric conversion capability of the particle to activate and regulate the neuron cell [175] (Reproduced with permission from de Ávila et al., Nature Communication; published by Springer Nature, 2017); (**c**) Correlation between propulsion speed of the mesoporous silica-based micromotors with a nanoswitch and pH of the surroundings [176] (Reproduced with permission from Patino et al., Nano Letters; published by American Chemical Society, 2019); (**d**) Image of propulsion mechanism and drug delivery of magnesium micromotors in the mouse stomach [177] (Reproduced with permission from Peng et al., Advanced Functional Materials; published by Wiley, 2017).

**Figure 8 micromachines-13-01473-f008:**
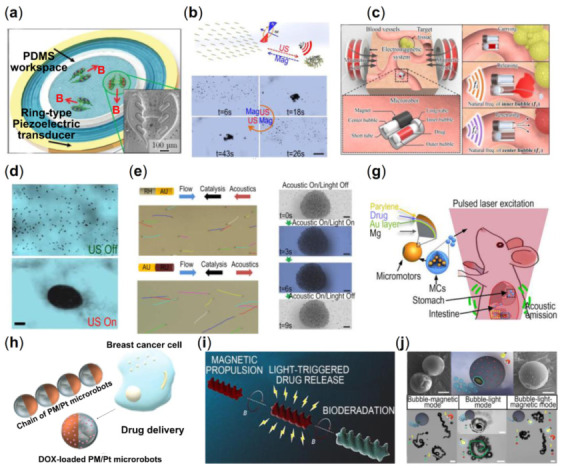
Schematic illustration of hybrid propulsion mechanism and applications. (**a**) Schematic diagram of magneto-acoustic actuation of CeFlowBots: An acoustic actuation test-bed comprising an annular piezoelectric transducer concentrically bonded to a glass substrate with a polydimethylsiloxane (PDMS)-based ring-shaped workspace on the opposite side [183] (Reproduced with permission from Mohanty et al., Small; published by Wiley, 2022); (**b**) Scheme of reversible assembly of the magnetic-acoustic hybrid nanomotors and snapshots of the directional swarm motion, aggregated nanomotor assembly at different times [184] (Reproduced with permission from Li et al., Nano Letters; published by American Chemical Society, 2015); (**c**) Schematic diagram of the designed microrobot and drug manipulation process(carrying, releasing, and penetrating) [61] (Reproduced with permission from Jeong et al., Sensors and Actuators A: Physical; published by Elsevier, 2015); (**d**) Microscopic images illustrating the swarming behavior of chemically powered Au−Pt nanomotors under the acoustic radiation forces (scale bar, 10 μm) [67] (Reproduced with permission from Xu et al., Journal of the American Chemical Society; published by American Chemical Society, 2015); (**e**) Directional collective motion of bimetallic micromotors (scale bar, 10 μm) [66] (Reproduced with permission from Ren et al., ACS Nano; published by American Chemical Society, 2017); (**f**) Time-lapse snapshots of behaviors of Au nanomotors (scale bar, 30 μm) [65] (Reproduced with permission from Zhou et al., Advanced Science; published by Wiley, 2018); (**g**) Schematic of the PAMR in the gastrointestinal tract. The micromotor capsules are administered into the mouse [185] (Reproduced with permission from Wu et al., Science Robotics; published by Amer Assoc Advancement Science, 2019); (**h**) Schematic diagram of the fabrication of PM/Pt microrobots for cell manipulation, anticancer doxorubicin (DOX) drug loading, and delivery [68] (Reproduced with permission from Villa et al., Advanced Functional Materials; published by Wiley, 2018); (**i**) Schematic diagram of the process of light-triggered drug release [70] (Reproduced with permission from Bozuyuk et al., ACS Nano; published by American Chemical Society, 2018); (**j**) Scanning-electron microscopy (SEM) images of the hybrid propulsion modes of microrobots and their track in hydrogen peroxide solutions [186] (Reproduced with permission from Yuan et al., Chemistry of Materials; published by American Chemical Society, 2020).

**Table 1 micromachines-13-01473-t001:** Examples of artificial micro/nanorobots and propulsion mechanisms.

Propulsion Mechanism	Micro Particles	Medium	Dimensions	Maximum Speed	Application	Reference
Magnetism	Zn-based artificial bacterial flagella	Tumor microenvironments	10 μm	50 μm/s	Tumor cell targeted drug delivery	[38]
	PEGDA based MNP	Blood	100 × 35 μm^2^	160 μm/s	Anticancer drug delivery	[39]
	Mushroom/red blood cell/teardrop/spherical-shaped child-parent microrobot	Acidic small intestinal environment	-	1000–5020 μm/s	Intestine-targeted therapy	[40]
	RBC-shaped Fe_3_O_4_ particle	Blood	≈2 μm	56.5 μm/s	Drug delivery	[41]
	Fe_3_O_4_@PDA nanoparticles	-	Diameter 400 nm	-	Ultrasound imaging contrast enhancement	[42]
	Fe_3_O_4_ nanoparticles	Ethylene glycol	Diameter 300–500 nm	2.6 mm/s	Microscale intestinal perforation patching	[43]
	Fe_3_O_4_ pine pollen-based nanoparticle	Gastrointestinal fluids	25 μm	175.19 μm/s	Targeted drug delivery	[44]
	Self-repelling collective NdFeB microrobots	Air-water interface	100–350 μm	42,000 μm/s	Adaptive navigation	[45]
Acoustic Field	Metallic microrods	Water	Length: 2 μm, Diameter: 300 nm	200 μm/s	-	[46]
	Nanoshell	Deionized water	0.5–5 μm	Depend on size	Biomedical delivery	[47]
	Microtubular structure with one end open	Water	Length: 450 μm/920 μm Height: 45 μm Width: 80 μm	1.35 mm/s	Drug delivery	[48]
	Microrocket with multilayer polymer covered by Pt nanoparticles	Hydrogen peroxide	Length: 10 μm Diameter: 1 μm	-	Micro-generator of electricity, drug delivery	[49]
	Bullet-shaped microrobot	Fluid medium	25 μm	2.2 mm/s	Targeted drug delivery	[50]
	Bimetallic head and a flexible tail	Aqueous solution	>10 μm	Depend on materials	In vivo application	[51]
Chemical energy driven	ZnO-Pt	<5% H_2_O_2_ solution	>5 μm	350 μm/s	Water-purification, Photocatalysts	[52]
	Multimetallic (Au/Ag/Ni/Au)	1.5% H_2_O_2_ solution	20 μm	Exceeding 209 μm/s	Biosensing	[53]
	PMMA-AgCl	Pure water (adding KNO_3_ to change conductivity)	2.5 μm	12 μm/s	Biomedicine, environmental remediation	[54]
	ZnO/SiO2	Pure water(air-exposed)	2.5 μm	7.8 μm/s	Biomedical and environmental remediation	[55]
	MSNP-SiO_2_	0~3% H_2_O_2_ solution	90 nm	-	Drug delivery	[56]
	Ur-PDA NC	In bladder	1 μm	10.67 μm/s	Intravesical therapeutic delivery	[57]
	TiO_2_-Pt	Pure water	800 nm	21 μm/s	Photocatalysts for organic pollutant remediation	[58]
	TiO_2_-Au	Pure water (0.1% H_2_O_2_)	~1.0 μm	25 μm/s	Environmental remediation	[59]
	TiO_2_-Fe	0~5% H_2_O_2_ solution	2 μm	Exceeding 11.65 μm/s	Carrying, identifying and separating in a complex environment	[60]
Acoustics and magnetism	Cylinder-shaped neodymium magnet with bubble	Water	900 μm	-	Targeted drug delivery	[61]
	Superparamagnetic particles	Vessel wall	1–2.9 μm	20 μm/s	Drug delivery	[62]
Acoustics and optic	Leukocyte membrane-coated gallium nanoswimmer	Blood	7.03 μm	108.7 μm/s	Photothermal chemical therapy	[63]
	Au/TiO_2_ microbowl	-	-	26.4 μm/s	Drug delivery	[64]
	Au/Pd/Ag/polypyrrole (PPy)/SiO_2_ nanomotor	-	≈40 μm	50 μm/s	Cargo transport, chemical sensing	[65]
Acoustics and chemistry	Rh-Au microrod	H_2_O_2_ solution	2–3 μm	-	Drug delivery	[66]
	Au-Pt nanomotor H_2_O_2_	H_2_O_2_ solution	2 μm	117.4 μm/s	Cargo transport, chemical sensing	[67]
Magnetism and chemistry	PM/Pt microrobots	2.5 wt% H_2_O_2_ cell culture	≈4.5 μm	2.0 μm/s	Capture and transportation of cancer cells	[68]
	SiO_2_ based acid-stable micropropeller	4% mucin solutions	500 nm	-	Targeted drug delivery in stomach	[69]
Magnetism and acoustics	SPION based Chitosan Microswimmers	Water	20 × 6 μm^2^	3.34 ± 0.71 μm/s	Minimally invasive surgery	[70]

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
