# Peer review of "Recent Process in Microrobots: From Propulsion to Swarming for Biomedical Applications"

_micromachines, 2022, doi:10.3390/mi13091473_

Round 1
Reviewer 1 Report
This paper presents a board review of microrobots that can be used in the variety of biomedical field. The authors summarized with the actuating mechanisms and the application, where this review is of great interest in the field of application in drug delivery, sophisticated surgery method, and medical therapies. Here are some minor comments:
1) Mechanisms in Figure 2e and f are hardly recognizable. Authors could include words for explanations or revise with other figures that are available.
2) Authors need to carefully revise all the figures. Some of the brackets are missing such as in Figure 3.
3) In chemical induced propulsion in Section 2.3.1, could authors comment how the actuators could collect hydrogen peroxide if the robot is inserted inside human body?
4) Also, for Figure 7, the various chemically induced actuating mechanisms have high potentials. How could these system achieve fuels in vivo?
Reviewer 2 Report
1. Recently, there are several review papers about the swarming of micro-robots such as:
- L. Yang, J. Yu, S. Yang et al., “A survey on swarm microrobotics,” IEEE Transactions on Robotics, 2021.
- L. Yang, and L. Zhang, “Motion Control in Magnetic Microrobotics: From Individual and Multiple Robots to Swarms,” Annual Review of Control, Robotics, and Autonomous Systems, vol. 4, no. 1, pp. 509-534, 2021.
In the introduction, please clarify what makes this review different from others.
2. Please enhance the quality of Figures
3. Why does the author select magnetic, acoustic field-driven, chemically driven, and hybrid-driven mechanisms for review? For this topic, optical and electric mechanisms would also be good choices. Thus, before you begin reviewing, please explain for this (the author can do this in the first paragraph of section 2 or the introduction section).
4. Make a space before citing References, as in: “... rotational motion [33,61]”->“... rotational motion [33,61]”.
5. There was not good connection between the review and the research in Table I, the author must make a better connection.
6. Please check all typos: such as “xm”-> xm…
7. In my opinion, “0” should be changed to “zero” in writing.
8. Although the paper title is: “Swarming of Microrobots and the Biomedical Applications”, the paper content did not focus on reviewing the swarming of microrobots and their applications. The single/general microrobot was discussed extensively in 2.1.1, while only a brief overview of swarming was discussed in 2.1.2. The author seems to focus more on all microrobots and their biomedical applications than only on swarming. Contents should be changed to match the paper title or inverted. The author also should mention the difference between multiple microrobot control and swarm microrobot control. Recent swarm microrobots research in 2022 should be added more.
9. The author should rearrange the content of each review in section 2 since it doesn't seem to be organized properly. For example, several paragraphs (“Magnetic actuation….[66]” in subsection 2.1 can be put in the basic propulsion mechanisms….

Round 2
Reviewer 2 Report
Several of my concerns have been addressed. However, I still have several comments as follows:
1. There is still room for improvement in the introduction part regarding motivation as well as the contribution of this review. You should try to improve further
2. The quality of the figures is still not good enough. There are some texts I cannot see. Please take another look at it.
3. Table I and the review (section 2) do not connect well. As an example, the authors listed many research articles in Table I, but the order of appearance in Section 2 does not follow Table I, causing a disconnect between the two. The author should rewrite it again.

Author Response
Response to Reviewer 2 Comments
(Round 2)
Point 1: There is still room for improvement in the introduction part regarding motivation as well as the contribution of this review. You should try to improve further.
Response 1: Thanks for your valuable comment. Actually, although various reviews have contributed to the development of microrobots[1-7], there are regrettably few comprehensive summaries of different mechanisms and discussions of the swarming behavior in biomedical application of microrobots. Therefore, we not only analyzed the mechanisms and methods of microrobot, but also focuses on how individual microrobots can accomplish collaborative work through interactive behaviors in biomedical application. Furthermore, the main distinction from the previous review is the inclusion and discussion of many latest articles related to biomedical applications in the past two years[8-18], which are partly listed below.
- Liu, C.; Xu, T.; Xu, L.-P.; Zhang, X. Controllable Swarming and Assembly of Micro/Nanomachines. Micromachines 2018, 9, 10, doi:10.3390/mi9010010.
- Riess, B.; Groetsch, R.K.; Boekhoven, J. The Design of Dissipative Molecular Assemblies Driven by Chemical Reaction Cycles. Chem 2020, 6, 552–578, doi:10.1016/j.chempr.2019.11.008.
- Wu, Z.; Chen, Y.; Mukasa, D.; Pak, O.; Gao, W. Medical micro/nanorobots in complex media. Soc. Rev. 2020, 49, 8088-8112. doi: 10.1039/d0cs00309c.
- Hu, M.; Ge, X.; Chen, X.; Mao, W.; Qian, X.; Yuan, W. Micro/Nanorobot: A Promising Targeted Drug Delivery System. Pharmaceutics 2020, 12, 665, doi: 10.3390/pharmaceutics12070665.
- Zhou, H.; Mayorga-Martinez, C.C.; Pané, S.; Zhang, L.; Pumera, M. Magnetically Driven Micro and Nanorobots. Rev. 2021, 121, 4999–5041, doi: 10.1021/acs.chemrev.0c01234.
- Wang, Q.; Zhang, L. External Power-Driven Microrobotic Swarm: From Fundamental Understanding to Imaging-Guided Delivery. ACS Nano 2021, 15, 149–174, doi:10.1021/acsnano.0c07753.
- Wang, Z.; Xu, Z.; Zhu, B.; Zhang, Y.; Lin, J.; Wu, Y.; Wu, D. Design, Fabrication and Application of Magnetically Actuated Micro/Nanorobots: A Review. Nanotechnology 2022, 33, 152001, doi:10.1088/1361-6528/ac43e6.
- Law, J.; Wang, X.; Luo, M.; Xin, L.; Du, X.; Dou, W.; Wang, T.; Shan, G.; Wang, Y.; Song, P.; et al. Microrobotic Swarms for Selective Embolization. Adv. 2022, 8, eabm5752, doi:10.1126/sciadv.abm5752.
- Wang, Q.; Yang, S.; Zhang, L. Magnetic Actuation of a Dynamically Reconfigurable Microswarm for Enhanced Ultrasound Imaging Contrast. IEEE/ASME Trans. Mechatron. 2022, 1–11, doi:10.1109/TMECH.2022.3151983.
- Yue, H.; Chang, X.; Liu, J.; Zhou, D.; Li, L. Wheel-like Magnetic-Driven Microswarm with a Band-Aid Imitation for Patching Up Microscale Intestinal Perforation. ACS Appl. Mater. Interfaces 2022, 14, 8743–8752, doi:10.1021/acsami.1c21352.
- Sun, M.; Chan, K.F.; Zhang, Z.; Wang, L.; Wang, Q.; Yang, S.; Chan, S.M.; Chiu, P.W.Y.; Sung, J.J.Y.; Zhang, L. Magnetic Microswarm and Fluoroscopy‐Guided Platform for Biofilm Eradication in Biliary Stents. Advanced Materials 2022, 2201888, doi:10.1002/adma.202201888.
- Wang, Q.; Du, X.; Jin, D.; Zhang, L. Real-Time Ultrasound Doppler Tracking and Autonomous Navigation of a Miniature Helical Robot for Accelerating Thrombolysis in Dynamic Blood Flow. ACS Nano 2022, 16, 604–616, doi:10.1021/acsnano.1c07830.
- Zhao, Y.; Xiong, H.; Li, Y.; Gao, W.; Hua, C.; Wu, J.; Fan, C.; Cai, X.; Zheng, Y. Magnetically Actuated Reactive Oxygen Species Scavenging Nano-Robots for Targeted Treatment. Advanced Intelligent Systems 2022, 4, 2200061, doi:10.1002/aisy.202200061.
- Sun, M.; Chan, K.F.; Zhang, Z.; Wang, L.; Wang, Q.; Yang, S.; Chan, S.M.; Chiu, P.W.Y.; Sung, J.J.Y.; Zhang, L. Magnetic Microswarm and Fluoroscopy-Guided Platform for Biofilm Eradication in Biliary Stents. Advanced Materials 2022, 2201888, doi:10.1002/adma.202201888.
- Peng, X.; Urso, M.; Ussia, M.; Pumera, M. Shape-Controlled Self-Assembly of Light-Powered Microrobots into Ordered Microchains for Cells Transport and Water Remediation. ACS Nano 2022, 16, 7615–7625, doi:10.1021/acsnano.1c11136.
- Ketzetzi, S.; Rinaldin, M.; Dröge, P.; Graaf, J. de; Kraft, D.J. Activity-Induced Interactions and Cooperation of Artificial Microswimmers in One-Dimensional Environments. Nat Commun 2022, 13, 1772, doi:10.1038/s41467-022-29430-1.
- Cheng, Y.; Mou, F.; Yang, M.; Liu, S.; Xu, L.; Luo, M.; Guan, J. Long-Range Hydrodynamic Communication among Synthetic Self-Propelled Micromotors. Cell Rep. Phys. Sci. 2022, 3, 100739, doi:10.1016/j.xcrp.2022.100739.
- Mohanty, S.; Paul, A.; Matos, P.M.; Zhang, J.; Sikorski, J.; Misra, S. CeFlowBot: A Biomimetic Flow‐Driven Microrobot That Navigates under Magneto‐Acoustic Fields. Small 2022, 18, 2105829, doi:10.1002/smll.202105829.
Point 2: The quality of the figures is still not good enough. There are some texts I cannot see. Please take another look at it..
Response 2: Thanks for your valuable comment. We are sorry that the unclear text of the figure in the previous manuscript confused the reviewer. We have improved the quality of the figure and its corresponding text: (1) the parts of the figure that do not affect comprehension have been removed; (2) the key parts of the figure have been enlarged; and (3) the font size on the figure has been increased to help make the text more clearly visible.
Point 3: Table I and the review (section 2) do not connect well. As an example, the authors listed many research articles in Table I, but the order of appearance in Section 2 does not follow Table I, causing a disconnect between the two. The author should rewrite again.
Response 3: We sincerely thank you for your constructive comment. After carefully studying the issues you mentioned, we find that in the previous manuscript, the literature listed in the Table I did not exactly match the order of appearance in the text. We have checked the text and rearranged the literature numbers in the table to match the order of appearance.
